# A role for condensin in mediating transcriptional adaptation to environmental stimuli

Lucy Lancaster[1], Harshil Patel[2] , Gavin Kelly[2] , Frank Uhlmann[1]

Nuclear organisation shapes gene regulation; however, the principles by which three-dimensional genome architecture influences gene transcription are incompletely understood. Condensin is a key architectural chromatin constituent, best known for its role in mitotic chromosome condensation. Yet at least a subset of condensin is bound to DNA throughout the cell cycle. Studies in various organisms have reported roles for condensin in transcriptional regulation, but no unifying mechanism has emerged. Here, we use rapid conditional condensin depletion in the budding yeast *Saccharomyces cerevisiae* to study its role in transcriptional regulation. We observe a large number of small gene expression changes, enriched at genes located close to condensin-binding sites, consistent with a possible local effect of condensin on gene expression. Furthermore, nascent RNA sequencing reveals that transcriptional down-regulation in response to environmental stimuli, in particular to heat shock, is subdued without condensin. Our results underscore the multitude by which an architectural chromosome constituent can affect gene regulation and suggest that condensin facilitates transcriptional reprogramming as part of adaptation to environmental changes.

## Introduction

Condensin is a key regulator of chromatin architecture which undergoes dynamic regulation during the cell cycle (Hirano, 2016; Uhlmann, 2016). Through promoting chromosome condensation during mitosis, condensin compacts DNA to form chromosomes, crucial for the successful segregation of genetic material amongst daughter cells. Between cell divisions, condensin remains detectable in the nucleus and has been implicated in various roles in interphase genome organisation and stability (Aono et al, 2002; Heale et al, 2006; Hartl et al, 2008; Kakui et al, 2020). Whether and how condensin impacts on gene expression in interphase and in mitosis remains incompletely understood.

Condensin is a large pentameric protein complex and has a ring structure comprising two structural maintenance of chromosome (SMC) complex subunits Smc2 and Smc4. Three additional subunits associate with the complex: Brn1, the kleisin subunit that bridges the Smc2 and Smc4 ATPase heads, and HEAT repeat subunits Ycg1 and Ycs4 (budding yeast subunit nomenclature is used throughout the article, if not indicated otherwise). All five subunits are essential for chromatin association of budding yeast condensin and for chromosome condensation (Lavoie et al, 2002). The budding yeast condensin complex is bound to DNA throughout the cell cycle, with condensin levels peaking around centromeres in mitosis. In addition to centromeres, condensin has been observed along chromosome arms. Binding sites are found at tRNA genes, promoters of ribosomal protein and other highly expressed genes where condensin overlaps with the cohesin loader complex, the rDNA, telomeres, heterochromatin, and sites of converging replication (Wang et al, 2005; D'Ambrosio et al, 2008; Lopez-Serra et al, 2014). There are two distinct condensin complexes in higher organisms which share the SMC subunits and vary in their additional subunits. Whereas condensin I becomes enriched on chromosomes after nuclear envelope breakdown in mitosis, condensin II remains nuclear throughout interphase (Hirota et al, 2004; Ono et al, 2004).

A variety of evidence across several organisms suggests a role for condensin in transcriptional regulation. Examples include condensin functions in both up- and down-regulation of gene expression. Early work in the fruit fly *Drosophila melanogaster* suggested a contribution of condensin to Polycomb group-mediated gene silencing (Lupo et al, 2001). Mouse condensin II, in turn, promotes histone gene expression by establishing interactions between histone gene clusters in embryonic stem cells (Yuen et al, 2017). In human breast cancer cells, both condensin I and II have been found to contribute to licensing estrogen receptor-dependent enhancer transcription and consequent gene activation (Li et al, 2015). In chicken DT40 cells, depletion of the condensin I kleisin subunit CAP-H causes widespread gene expression changes (Zhang et al, 2016), whereas in the plant *Arabidopsis thaliana*, mutation of the condensin subunit Smc4 leads to de-repression of transposons within pericentromeric chromatin (Wang et al, 2017). In the budding yeast *Saccharomyces cerevisiae*, condensin contributes to compacting chromatin when cells enter quiescence, thereby reducing chromatin accessibility to the transcriptional machinery and promoting transcriptional silencing (Wang et al, 2016; Swygert et al, 2019).

[1]Chromosome Segregation Laboratory, The Francis Crick Institute, London, UK    [2]Bioinformatics and Biostatistics Science Technology Platform, The Francis Crick Institute, London, UK

Correspondence: frank.uhlmann@crick.ac.uk

At the budding yeast rDNA locus, condensin was shown to contribute to transcriptional silencing in conjunction with the histone deacetylase Sir2 (Machin et al, 2004).

Consistent with roles in regulating gene expression, condensin has been reported to directly interact with transcription factors. Budding yeast condensin is found at RNA polymerase (pol) III–transcribed genes, where it interacts with the RNA pol III transcription factor TFIIIC (D'Ambrosio et al, 2008; Haeusler et al, 2008). In the fission yeast Schizosaccharomyces pombe, condensin is recruited to both RNA pol III, as well as highly transcribed RNA pol II genes, by the general transcription factor TATA box–binding protein (TBP) (Iwasaki et al, 2015). Additional cell cycle stage-specific transcription factors promote condensin-dependent chromatin interactions (Kim et al, 2016), whereas DNA access for condensin is provided by the Gcn5 histone acetyl transferase together with the RSC chromatin remodeller (Toselli-Mollereau et al, 2016; Muñoz et al, 2020). Condensin interaction with TFIIIC was also observed in vertebrates, where condensin is also found at promoters of actively expressed genes and in addition interacts with the epigenetic H3K4$^{me3}$ mark (Sutani et al, 2015; Yuen et al, 2017).

When assessing the role of condensin on gene expression, it is important to consider indirect effects from compromised condensin function on cell cycle progression, notably chromosome mis-segregation during cell divisions. Gene expression changes that were observed after condensin inactivation in asynchronously growing S. pombe cells were consequently attributed to unequal inheritance of the RNA exosome (Hocquet et al, 2018). Condensin mutation in a T-cell lymphoma, in turn, caused gene expression changes consistent with those expected to arise from aneuploidy (Woodward et al, 2016). Condensin II depletion in post-mitotic mouse hepatocytes revealed little evidence for gene expression changes (Abdennur et al, 2018 Preprint), whereas little impact on global gene expression was reported after condensin depletion in asynchronously growing S. cerevisiae cells (Paul et al, 2018). These considerations emphasise the need for fast condensin depletion, as well as control over cell cycle progression, to avoid indirect effects when studying condenin's role in gene expression.

# Results

### A transcriptional response to condensin depletion

To investigate the impact of condensin on gene expression in budding yeast, we constructed a strain to rapidly shut off condensin function within a single cell cycle phase. The condensin subunit Ycg1 was selected for depletion as its levels are naturally cell cycle regulated and appear to be a limiting factor of condensin activity (Doughty et al, 2016). The shut off strain (ycg1$^{Degron1}$) was attained by replacing the YCG1 promoter with the MET3 promoter that allows transcriptional repression in the presence of methionine. Ycg1 was additionally fused to three repeats of a minimal auxin-inducible degron tag (AID*) that targets the subunit for degradation in the presence of auxin and the Tir1 F-box protein (Nishimura et al, 2009; Morawska & Ulrich, 2013). A Pk epitope was included in the degron tag to facilitate detection. We propagated cells in synthetic medium lacking methionine to maintain

Ycg1 expression, then arrested cells in the G1 phase of the cell cycle by pheromone α-factor treatment. 2 h after a shift to rich medium (containing methionine) and auxin addition, Ycg1 was undetectable by Western blotting. Control cells harboured a Pk epitope fusion to Ycg1 as the sole modification of the YCG1 locus and were otherwise treated the same. Ycg1 depletion led to a pronounced ribosomal DNA segregation defect in the first cell division following release from the α-factor block, confirming that condensin had been functionally inactivated. Consequently, ycg1$^{Degron1}$ cells were inviable on rich medium containing auxin (Fig S1A–C).

To analyse the condensin contribution to gene regulation, we repeated the cell synchronisation experiment and collected samples of control and ycg1$^{Degron1}$ cells 2 h following depletion in the α-factor block (G1 sample). We then released cells to transition from G1 through S phase into a nocodazole-imposed mitotic arrest where we took another sample (M sample) (Fig 1A). RNA was then extracted. One half of the RNA sample was analysed by total RNA sequencing. From the other half, poly A tail-containing mRNAs were enriched before sequencing to specifically study the expression of protein-coding genes. Three biological repeats of the experiment were performed. When analysing the principle components of differences between the samples, the greatest difference stemmed from the G1 or M cell cycle state of the samples, consistent with expected cell cycle-dependent gene regulation (Fig S1D) (Spellman et al, 1998). The second principal component separated the control and ycg1$^{Degron1}$ samples, suggesting that the presence or absence of condensin has an impact on gene expression. In contrast, the repeat experiments and the two different RNA preparation methods clustered close together, confirming reproducibility of the gene expression profiles.

We applied a 1.5-fold threshold to identify genes that were differentially expressed between the control and ycg1$^{Degron1}$ strains. mRNA sequencing of the G1 sample revealed higher expression of ~300 genes in the strain depleted for condensin, when compared with the control strain, whereas 150 had lower expression (Fig 1B). Gene expression differences were also seen between the M samples, albeit at somewhat smaller numbers. As expected from the lower sequencing depth of the total RNA sample, we identified fewer differentially expressed genes in this dataset, most of which overlapped with those identified by mRNA sequencing (Fig S1E). When we look at the gene ontology of genes that showed elevated expression in condensin-depleted cells, we find numerous gene families involved in metabolism. Conversely, down-regulated genes often had functions in DNA maintenance and DNA repair. We will discuss possible reasons behind these gene ontologies, below.

As condensin is a structural chromatin component, we asked whether the gene expression differences coincided with specific chromosomal regions. We used a distance-based analysis to determine whether differentially expressed genes are closer to certain chromosomal features than expected by chance. A value of <1 in this analysis indicates that differentially expressed genes are spatially close to a feature, whereas values >1 indicate that fewer differentially expressed genes than expected by chance surround the feature. This analysis revealed a strong correlation between differentially expressed genes in G1 and condensin-binding sites (Fig 1C) (D'Ambrosio et al, 2008). Many of the affected genes close to condensin-binding sites were identified as ribosomal protein genes, which were consistently found to be expressed at a lower

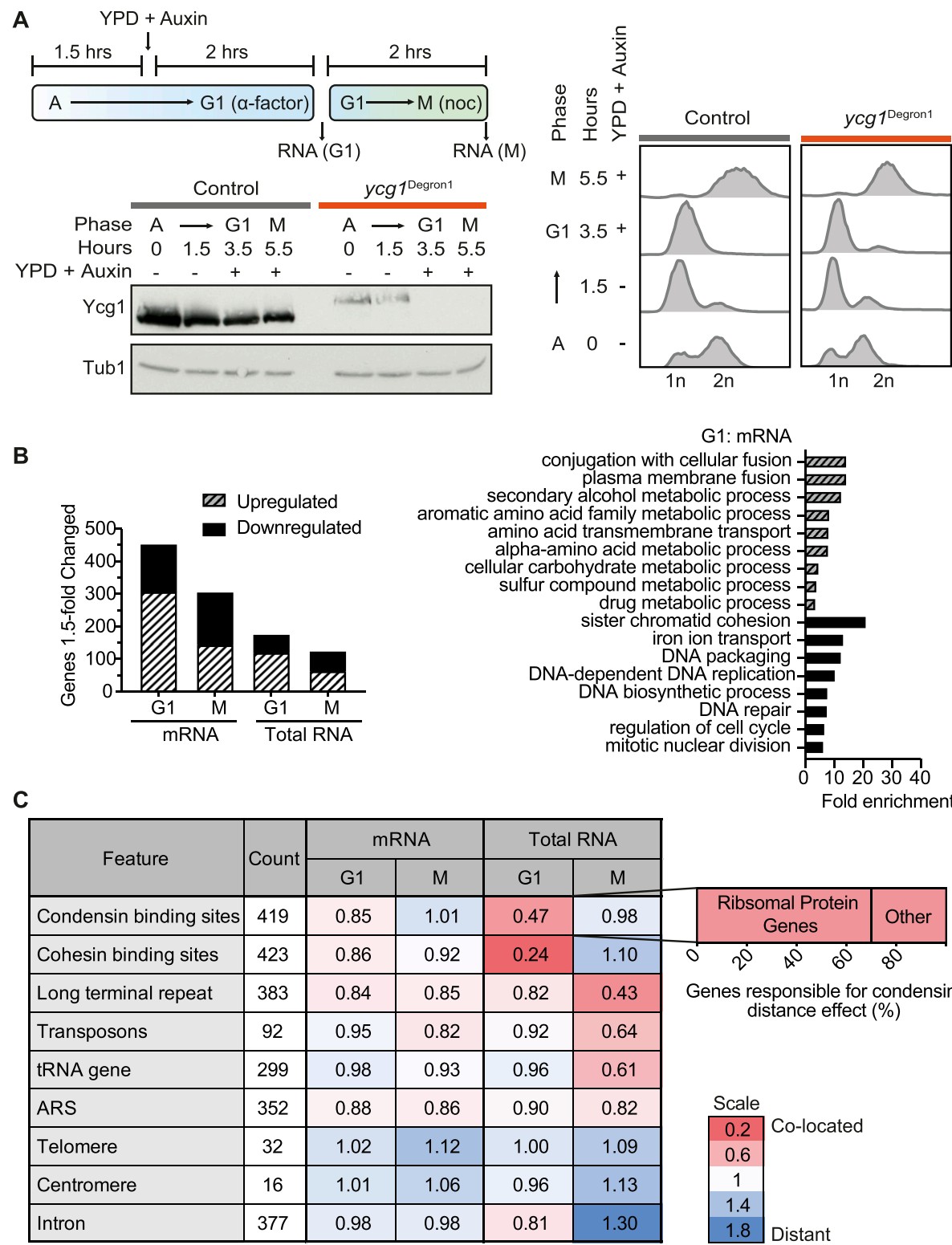

**Figure 1. The transcriptional response to condensin depletion.**
**(A)** Schematic of the cell synchronisation approach. **(A)** Asynchronous (A) control and *ycg1*^Degron1 cells were grown in minimal media and arrested in G1 by α-factor treatment for 1.5 h before transfer to rich YPD medium containing auxin to deplete Ycg1 for 2 h, when the G1 sample was taken. Cells were released into nocodazole-containing medium for 2 h to achieve a mitotic arrest, when a second sample was taken. Cell cycle progression was monitored by flow cytometry analysis of DNA content. Protein extracts were prepared for Western blotting against Ycg1-Pk in the control and Ycg1-Pk-3AID* in the *ycg1*^Degron1 strain. α-tubulin served as a loading control. **(B)** Differential gene expression analysis performed on mRNA and total RNA sequencing samples from three independent experiments, comparing the control and *ycg1*^Degron1

level (Fig S2A). Another notable feature that correlated with differentially expressed genes were cohesin-binding sites (Ocampo-Hafalla et al, 2007), although we do not at present have an explanation for this correlation. Other chromosomal features that are present in comparable numbers, for example, origins of DNA replication or intron-containing genes, did not correlate with differential gene expression. The localisation association was more pronounced when looking at the total RNA sequencing results, than the mRNA sequencing data. This could be due to the smaller number of significantly changed genes identified in the former analysis, which could have increased the association score of the most strongly affected genes. Taken together, this analysis suggests that condensin exerts a local effect on gene expression in G1. We will see below that passage through S phase in the absence of condensin leads to gene expression changes that are an indirect consequence of condensin inactivation, making the similar distance analysis of differentially expressed genes in the M sample less meaningful.

We confirmed that gene expression differences seen in our mRNA sequencing data could be independently reproduced by quantitative real-time PCR (qPCR) analysis. Fig 2A shows an example of the pheromone response gene *FIG1*, which showed higher expression in condensin-depleted G1 cells.

Our transcription analysis compared control cells with *ycg1*<sup>Degron1</sup> cells after depletion treatment. A confounding factor arises from the fact that condensin levels were lower in *ycg1*<sup>Degron1</sup> cells already before depletion (Fig 1A). Although the fidelity of rDNA segregation, a sensitive readout for condensin function, was unaffected before depletion (Fig S1C), we cannot exclude that gene expression differences existed between the control and *ycg1*<sup>Degron1</sup> strain already before condensin depletion.

### rRNA expression is elevated after condensin depletion

We next analysed transcription at the rDNA locus that is enriched for condensin and whose histone acetylation state is thought to be regulated by condensin (Machin et al, 2004). As rDNA transcripts have very long half-lives, we focussed our analysis on an intronic region, *ITS1*, within the RNA polymerase I–transcribed 35S rDNA gene (Fig 2B). This intron sequence is short-lived and provides a readout for ongoing 35S rRNA expression. Total RNA sequencing, as well as qPCR analysis, showed that *ITS1* expression was unaffected by condensin depletion in G1 (Figs 2C and S2B). In the mitotic sample, *ITS1* expression was slightly but significantly elevated in the *ycg1*<sup>Degron1</sup> strain compared with the control. This suggests that condensin-dependent epigenetic changes at the rDNA have a small impact on rRNA transcription.

rDNA copy number in budding yeast is sensitive to perturbations (Saka et al, 2016). Because condensin is a prominent rDNA-binding protein, we wanted to examine rDNA copy number stability in our *ycg1*<sup>Degron1</sup> strain. We measured the size of chromosome XII, harbouring the rDNA repeats, using pulsed-field gel electrophoresis. This revealed that merely fusing a Pk epitope tag to Ycg1 caused an

rDNA copy number reduction, compared with our otherwise isogenic wild-type strain (Fig S2C). The *ycg1*<sup>Degron1</sup> strain displayed a further reduction. This suggests that rDNA repeat stability is sensitive to perturbations in the condensin complex. Given the only slight changes to rRNA expression that we observed between our experimental strains, this emphasises the robustness of rRNA expression homeostasis over a wide range of rDNA repeat numbers (French et al, 2003), a mechanism that appears to operate largely independently of condensin. We cannot exclude that the rDNA copy number change has indirectly contributed to gene expression differences observed between the control and *ycg1*<sup>Degron1</sup> strains.

### Changes to histone gene expression after condensin depletion

Condensin has been implicated in histone gene regulation in both fission yeast and vertebrates (Kim et al, 2016; Yuen et al, 2017). We therefore assessed histone gene expression in our mRNA sequencing data. This revealed strikingly elevated mRNA levels at five out of the eight budding yeast histone genes in *ycg1*<sup>Degron1</sup> cells compared with the control (Figs 3A and S3A). However, elevated expression was only observed in our M but not the G1 sample. To corroborate these observations, we repeated mRNA measurements for four of these histone genes using qPCR. This confirmed elevated histone gene expression in M but not G1 cells depleted of condensin (Figs 3B and S3B). To understand the reason for the striking cell cycle dependence of differential histone gene expression, we gave further consideration to our experimental design. The cell synchronisation protocol used for both our G1 and M sequencing samples seeks to avoid indirect effects from cell division without condensin. However, cells in our M sample passed through S phase with reduced condensin levels. We therefore decided to assess whether passage through the S phase after condensin depletion might be a reason for increased histone gene expression.

We again synchronised *ycg1*<sup>Degron1</sup> cells in G1 by α-factor treatment, but this time we released cells to pass through S phase into nocodazole-imposed mitotic arrest in synthetic minimal medium lacking methionine to maintain *YCG1* expression. Only once all cells had completed S phase, the medium was exchanged for rich medium containing methionine and auxin. We then maintained the mitotic arrest for an additional 2 h, after which Ycg1 had become undetectable by Western blotting (Fig S3C). Histone gene expression, measured by qPCR for seven histone gene transcripts in three biological repeats of the experiment, now remained unaffected by condensin depletion (Figs 3C and S3D). This suggests that the up-regulation of histone gene expression observed in our original M samples arose indirectly from cell cycle progression through S phase without condensin. Condensin accumulates at and facilitates recovery of stalled DNA replication forks (Aono et al, 2002; D'Ambrosio et al, 2008). Therefore, increased histone gene expression after condensin depletion could be the result of genome damage that arose during DNA replication. Alternatively, condensin might contribute to dampening histone gene expression during S phase exit, after their up-regulation during DNA replication.

strains in both G1 and mitotic. Gene ontology analysis for overrepresentation was performed on >1.5-fold up- or down-regulated genes using the Panther tool (Thomas et al, 2006). (C) Spatial correlation analysis between differentially expressed genes and genome features. For each differentially expressed transcript, we found the distance to the nearest feature of those under consideration. We then formed the ratio of the median of those distances compared with the median distance of all not differentially expressed genes. The identity of genes contributing to the enrichment effect around condensin-binding sites is displayed in the accompanying graph.

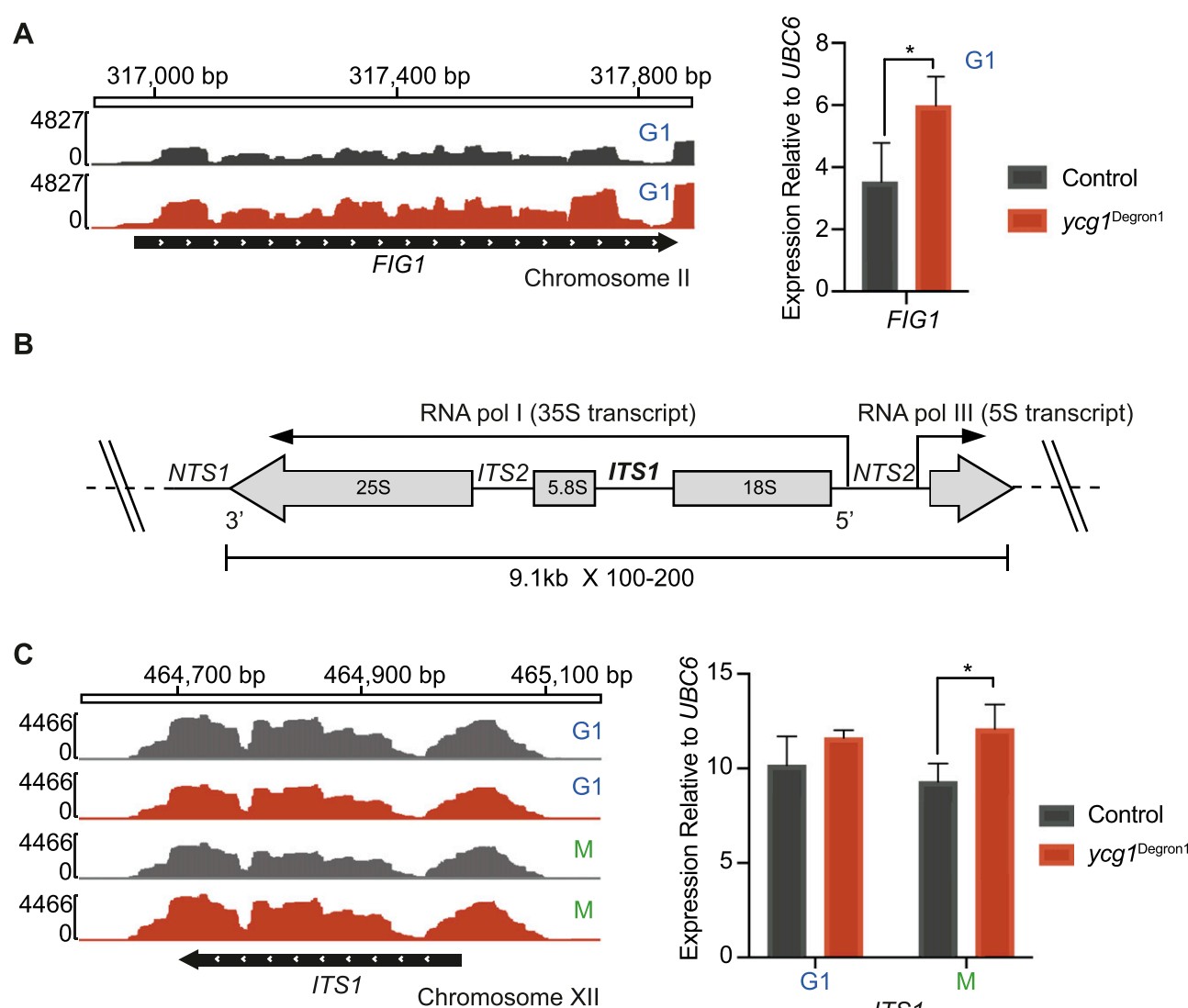

**Figure 2. rRNA expression is slightly elevated after condensin depletion.**
**(A)** *FIG1* mRNA levels in in the control and *ycg1*[Degron1] strain in G1 visualised with the Integrative Genomics Viewer (Robinson et al, 2011). *FIG1* mRNA levels were also assessed by qPCR, relative to *UBC6*. The means ± standard deviations from three independent experiments are shown, *($P$ = 0.05, unpaired $t$ test). *UBC6* transcript levels were similar between the two strains (Fig S2B). **(B)** Structure of an rDNA repeat, illustrating the position of the *ITS1* intron within the 35S rRNA transcript. **(C)** *ITS1* expression in G1 and mitotic. Total RNA reads were filtered to isolate reads overlapping *ITS1*. *ITS1* expression was also analysed by qPCR, relative to *UBC6*, the means ± standard deviations from three independent experiments are shown. *($P$ = 0.015, unpaired $t$ test). *ITS1* and *UBC6* transcript levels before normalisation can be found in Fig S2B.

## Transposon gene regulation by condensin

Transposons were amongst the affected genes in our differential gene expression analysis comparing the *ycg1*[Degron1] and control strains. Of the transposon classes in budding yeast, in particular Ty2 transposons showed reduced expression in cells depleted of condensin (Figs 4A and S4A). The effect was most pronounced in our G1 samples but was also observed in mitosis. As the transposon *GAG* gene shows the greatest divergence between transposon classes, we used three primer pairs to this gene to assess expression levels by qPCR. This analysis confirmed that indeed Ty2, but not Ty1, expression was almost twofold lower in condensin-depleted G1 cells when than in control cells (Figs 4B and S4A).

Transposons are mobile genetic elements that facilitate genome recombination and evolution (Chénais et al, 2012). At times of stability, there is little evolutionary advantage in transposon expression. However, during periods of environmental change, transposon expression can provide an adaptive advantage, for example, through new insertions that alter gene expression. To achieve condensin depletion by *MET3* promoter repression, we transitioned cells from synthetic minimal medium to rich medium containing methionine. Such a medium change is sensed by cells as a stress signal, which might induce transposon expression (Gasch et al, 2000; Türkel et al, 2009). It is therefore possible that condensin plays a role in regulating transposon transcription in response to the medium change. To investigate this possibility, we altered our experimental

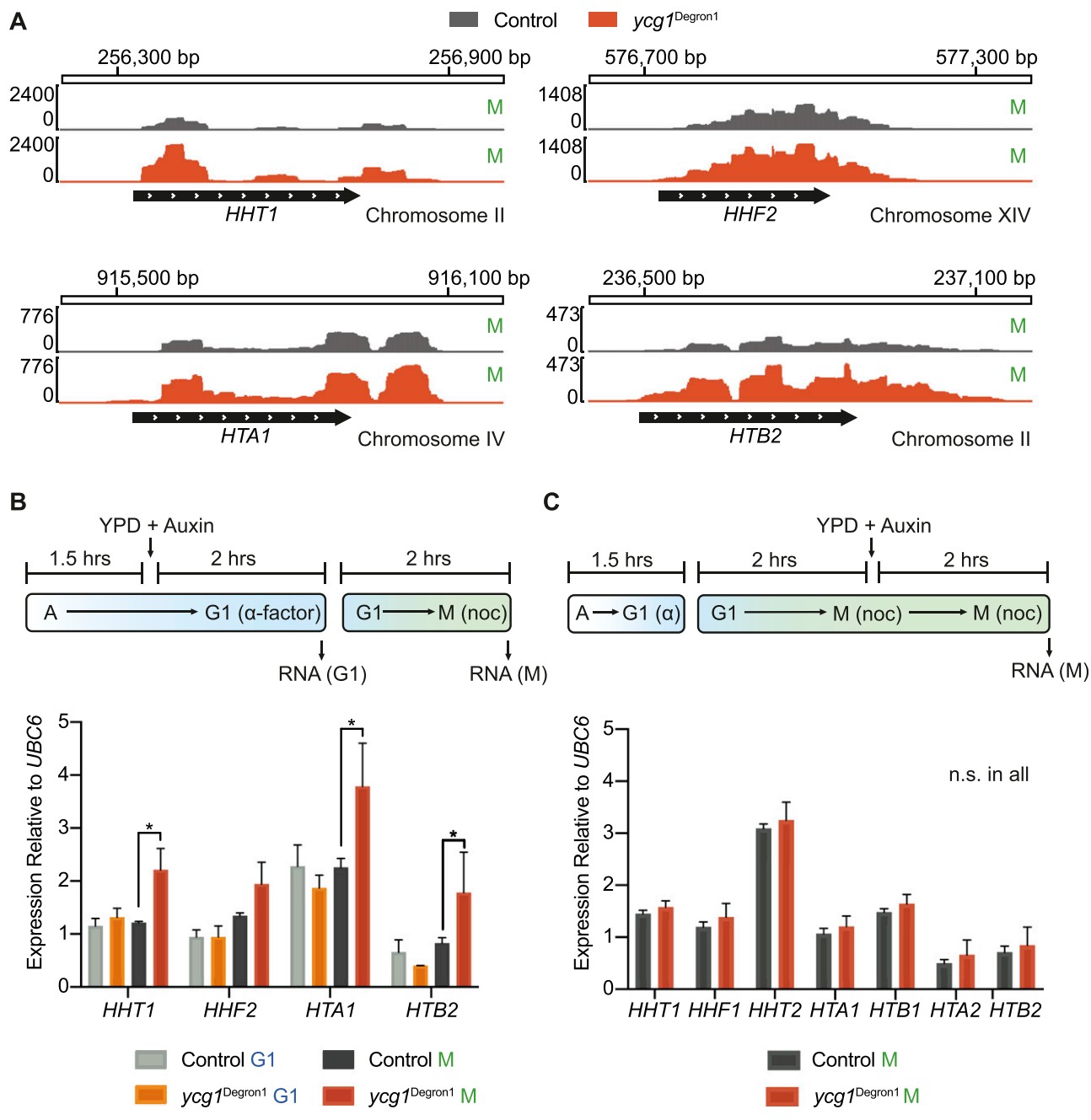

**Figure 3. Histone gene expression in response to condensin depletion.**
**(A)** Histone gene *HHT1*, *HHF2*, *HTA1*, and *HTB2* mRNA levels in the control and *ycg1*[Degron1] strains in mitosis. **(B)** Schematic of condensin depletion in G1 before progression through S phase in the absence of condensin. mRNA levels of the indicated histone genes was measured by qPCR relative to *UBC6*. The means ± standard deviations from three independent experiments are shown. *(P = 0.0161, 0.0008, and 0.0196 in case of *HHT1*, *HTA1*, and *HTB2*, respectively, unpaired *t* test). **(C)** Schematic of condensin depletion in mitosis. mRNA levels of the indicated histone genes was measured by qPCR relative to *UBC6*. The means ± standard deviations from three independent experiments are shown. n.s., no significant difference in a *t* test. Histone gene and *UBC6* transcript levels before normalisation can be found in Fig S3B and D.

protocol. After G1 arrest in synthetic minimal medium using an α-factor, instead of transitioning the cells to rich medium, we simply added methionine and auxin to achieve *MET3* promoter repression and protein degradation. After 2 h, Ycg1 was again undetectable (Fig S4B). Now, no difference in Ty2 transposon expression was observed (Figs 4C and S4A). Together, these results open the possibility that condensin contributes to the

up-regulation of Ty2 transposon expression following an environmental change to the growth medium.

### Nascent RNA analysis after condensin depletion

The above analysis suggested that, rather than maintaining steady state gene transcription, a role of condensin might lie in facilitating

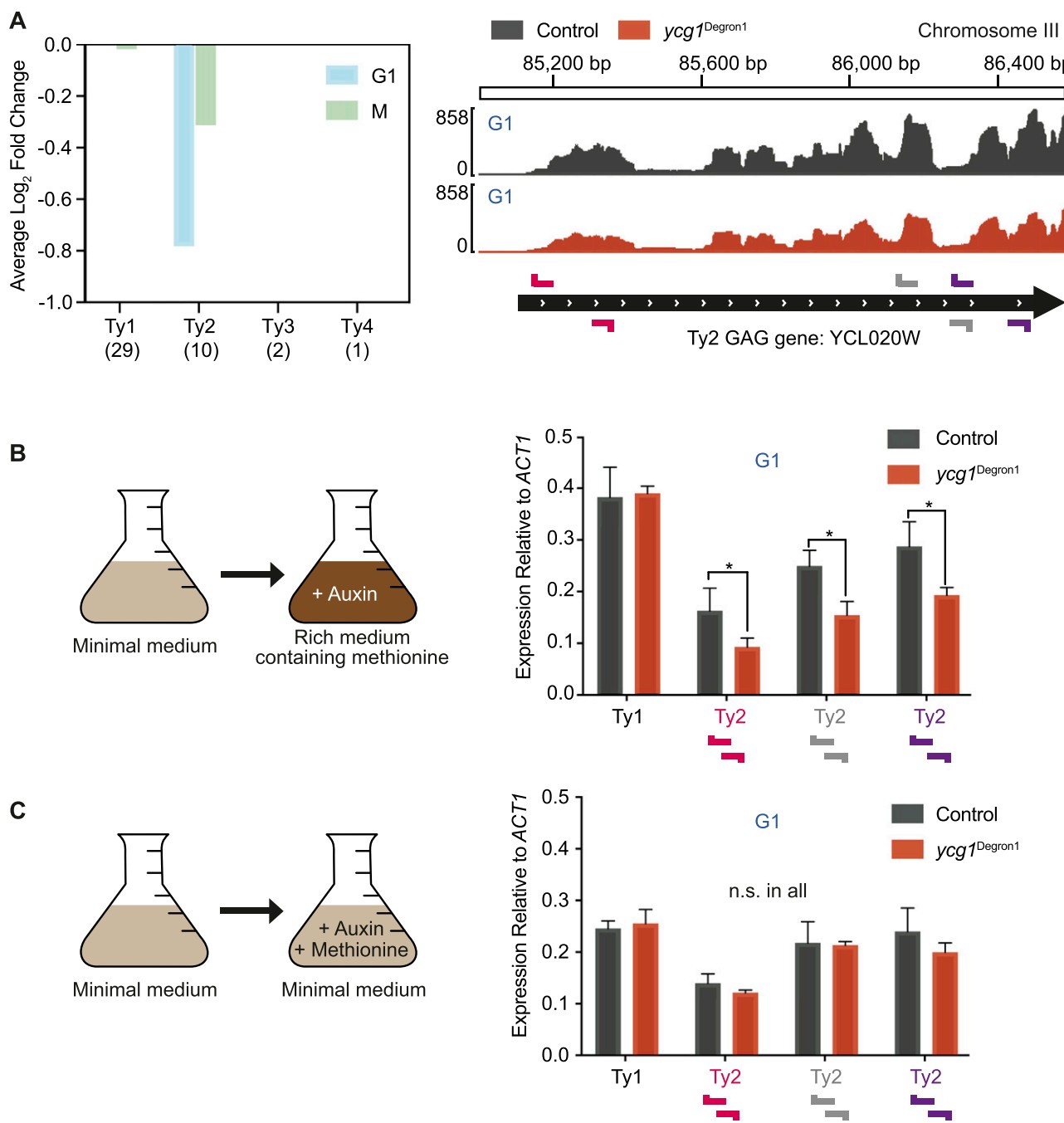

**Figure 4. Transposon gene expression in response to condensin depletion.**
**(A)** Differential expression analysis of the *GAG* transposon genes by transposon class in G1 and mitosis. Repetitive read numbers were divided equally between mappable locations, and the average fold change across all full *GAG* transposon genes by class is shown. Numbers denote the number of full *GAG* genes annotated in the *Saccharomyces cerevisiae* genome database. Expression of the Ty2 *GAG* gene *YCL020w* in G1 is shown in control and *ycg1*Degron1 strains. The locations of primer pairs for qPCR analysis are shown. **(B)** qPCR analysis of Ty1 and Ty2 transposon *GAG* gene mRNA levels, relative to *ACT1*. RNA was extracted in G1-arrested cells following shift from minimal synthetic to rich (YPD) medium. Means ± standard deviations from three independent experiments are shown. *(*P* = 0.0272, 0.0046 and 0.0049 for the three primer pairs from left to right, unpaired *t* test). **(C)** qPCR analysis of Ty1 and Ty2 transposon *GAG* gene mRNA levels, where cells were maintained in minimal synthetic medium and condensin depletion was achieved by methionine and auxin addition. Means ± standard deviations from three independent experiments are shown. n.s., no significant difference in an unpaired *t* test. *GAG* gene and *ACT1* transcript levels before normalisation can be found in Fig S4A.

gene expression changes in response to external stimuli. To explore this possibility further, we turned to the well characterised cellular response to heat shock. Before we analysed the transcriptional response to heat shock, we introduced two changes to our experimental strategy. First, we

used an alternative allele, *ycg1*Degon2, to deplete Ycg1. *ycg1*Degon2 includes only one instead of three AID* tag repeats (Morawska & Ulrich, 2013). This resulted in greater Ycg1 stability before depletion, while allowing fast Ycg1 removal within 30 min after methionine and auxin addition (Fig 5A). Again,

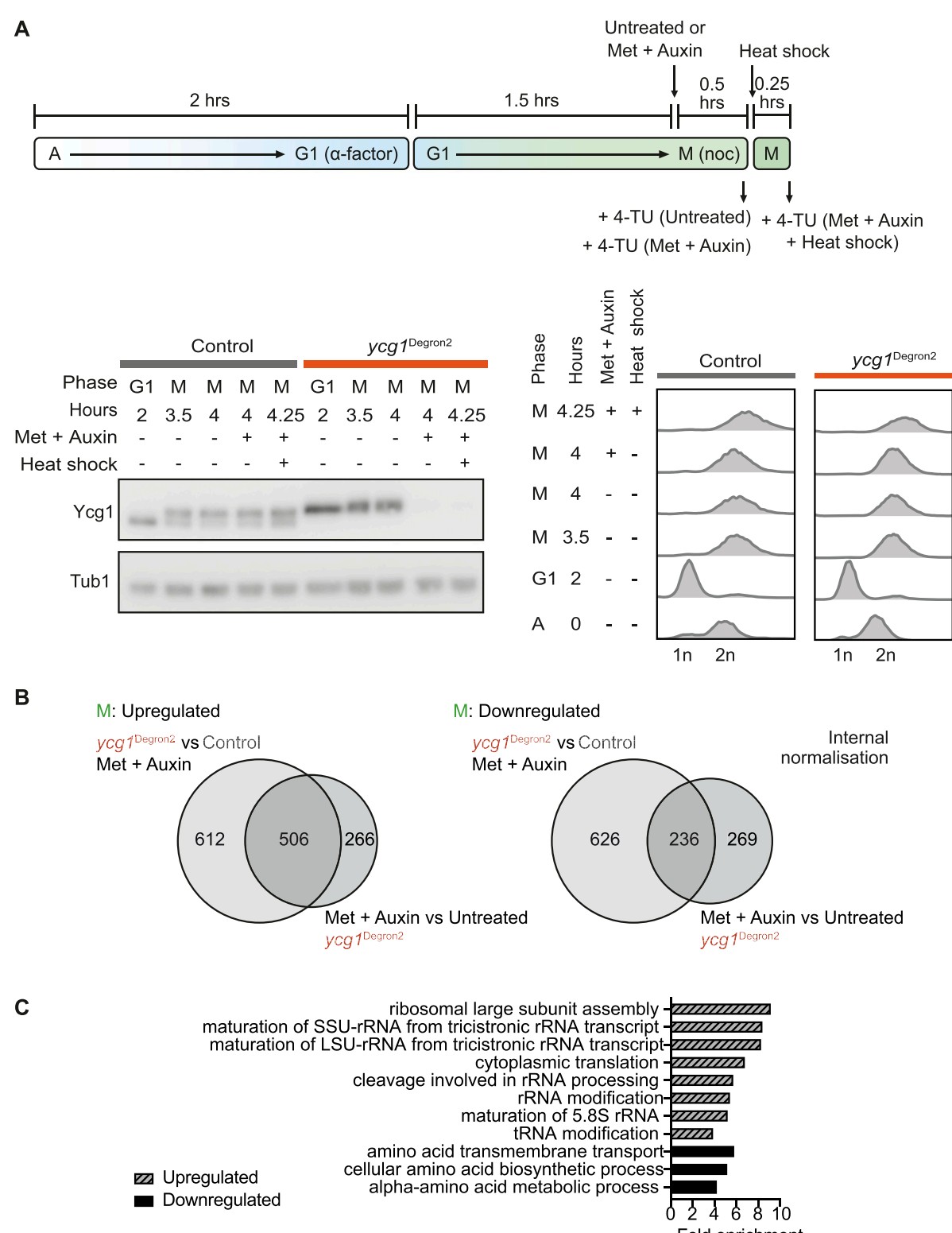

**Figure 5. Nascent RNA analysis after condensin depletion.**
**(A)** Schematic of cell synchronisation and condensin depletion for TT$_{chem}$-seq sample collection. Cells were synchronised in G1 and released into nocodazole-imposed mitotic arrest for 1.5 h before methionine and auxin were added to two thirds of the culture. 4-TU pulses were applied after 30 min and RNA samples collected. Remaining cultures were heat-shocked for 15 min, pulsed with 4-TU and the final samples taken. Cell synchrony was confirmed by flow cytometric analysis of DNA content, Ycg1 depletion was confirmed by Western blotting. **(B)** Identification of genes differentially expressed as the result of condensin depletion. Three biological replicates were combined for the analysis. The Venn diagrams compare significantly up- or down-regulated genes when comparing either the control and *ycg1*$^{Degron2}$ strains after

rDNA segregation fidelity in $ycg1^{Degon2}$ cells, as a readout for condensin function, was comparable with control cells before, but markedly compromised after depletion (Fig S1C). Secondly, to better capture transcriptional changes, we adapted transient transcriptome sequencing ($TT_{chem}$-seq) (Gregersen et al, 2020) for use in budding yeast. In this approach, we pulse label newly synthesised RNA for 5 min with 4-thiouracil (4-TU). Nascent, label-containing RNA is then tagged with a thiol-reactive biotin derivative, purified, chemically fragmented and sequenced. 4-TU labelled fission yeast RNA was used as a spike-in to allow quantitative comparisons.

After synchronisation in G1 in the synthetic medium, we released $ycg1^{Degon2}$ and control cells into nocodazole-imposed mitotic arrest. When all cells had completed DNA replication after 90 min, methionine and auxin were added to half of the cultures for 30 min, whereas the other half was further incubated without additions. Samples were now taken from all four cultures by adding 4-TU for 5 min before harvest. Two further methionine and auxin treated $ycg1^{Degon2}$ and control cultures were then heat-shocked by transfer to 37°C for 15 min, before 4-TU addition and sampling (Fig 5A). Three biological repeats of the experiment were performed. Principle component analysis confirmed reproducibility between the repeats (Fig S5A).

We first used internal normalisation between the experimental conditions to identify differentially expressed genes after condensin depletion. We did this in two ways. First, we compared samples from the $ycg1^{Degron2}$ strain, depleted for condensin or not. Second, we compared the control and $ycg1^{Degron2}$ strains both treated with methionine and auxin. The first comparison revealed 772 genes whose expression was more than twofold up-regulated after condensin depletion, 506 of which were also identified in the second comparison (Fig 5B). In turn, 505 genes were found down-regulated by more than twofold in the first comparison of which 236 were also found in the second (a list of these genes can be found in Table S1). The control and $ycg1^{Degron2}$ strains showed gene expression differences already under untreated conditions (Fig S5A). Despite this, the highly significant overlap of gene expression differences when comparing the degron strain before and after depletion, or when comparing the control and degron strains under depletion conditions, lends confidence to the identification of condensin-regulated genes. Looking at the gene ontologies, we find genes involved in rRNA, ribosome and tRNA biogenesis up-regulated, whereas aspects of amino acid metabolism were down-regulated after condensin depletion (Fig 5C).

We also compared differentially expressed genes from our initial mRNA sequencing experiment, between the control and $ycg1^{Degron1}$ strains, with those identified by nascent RNA sequencing between the control and $ycg1^{Degron2}$ strains. 156 of 392 significantly up-regulated genes and 136 of 476 down-regulated genes in the first comparison were also recovered in the second comparison (Fig S5B, a gene list can be found in Table S1). Whereas this overlap is very significant, differences between the experiments, including the medium shift and passage through the S phase after condensin depletion in the mRNA but not the nascent RNA sequencing experiment, are a likely source for variation.

## Condensin promotes global transcriptional down-regulation in response to heat shock

After methionine and auxin treatment, control and $ycg1^{Degron2}$ cells were exposed to heat shock. We first compared the consequent gene expression changes in our wild-type strain, internally normalised, with a published microarray analysis of mRNA level changes after a comparable heat shock treatment (Gasch et al, 2000). 273 of 443 upregulated and 286 of 333 down-regulated genes in this study were found up- or down-regulated in our dataset (Fig S5C). We conclude that the transcriptional response to heat shock is adequately captured in our $TT_{chem}$-seq approach.

We next analysed absolute nascent RNA levels throughout our experiment by normalising read counts to the *S. pombe* 4-TU RNA spike-ins. We visualised transcription levels through a heat map of all genes, as well as global averaged metagene profiles (Fig 6A). This revealed that, overall, nascent transcription of control and $ycg1^{Degron2}$ cells before any treatment was comparable. This changed after methionine and auxin treatment, when transcription in control cells appeared somewhat attenuated, an effect that was not seen in the condensin depletion strain where transcription remained largely unaffected. An impact of the condensin status on nascent transcription became even more noticeable after the heat shock. The temperature shift resulted in prominent global down-regulation of gene expression in control cells, which was less pronounced in the $ycg1^{Degron2}$ strain.

To quantify the above effects on global transcription, we depicted the fraction of all mapped sequencing reads attributed to *S. cerevisiae* mRNA synthesis relative to spike-in *S. pombe* reads. This analysis confirmed a reduction (median 25% between the three replicates) of transcription in control cells after methionine and auxin addition (Fig 6B). In contrast, transcription slightly increased following the same treatment in the $ycg1^{Degron2}$ strain (median 4%). Heat shock resulted in a strong median 2.6-fold transcription repression in controls cells (over 3.5-fold compared with the untreated condition). Gene repression in response to heat shock was less pronounced in condensin-depleted cells, reaching a median 1.6-fold when comparing the sample from before with the sample after heat shock. This suggests that condensin plays a role in global transcriptional down-regulation after heat shock.

We then assessed, based on the spike-in normalisation, differentially expressed genes between $ycg1^{Degron2}$ cells and the control. This revealed only few differences in the untreated cells, but a substantial number of genes whose expression levels were more than twofold greater in $ycg1^{Degron2}$ cells after methionine and auxin addition (Fig 6C). After heat shock, more than half of all budding yeast genes were more than twofold more highly expressed in the absence of condensin than in its presence.

Heat shock is often thought of eliciting a characteristic transcriptional programme that involves up-regulation of heat shock response genes. Our analysis in contrast suggests that heat shock results in global transcriptional down-regulation, to which condensin contributes. This conclusion is consistent with a report of

treatment with methionine and auxin or the $ycg1^{Degron2}$ strain without and with methionine and auxin treatment ($P$ = 3.3 × 10$^{-221}$ and $P$ = 2.5 × 10$^{-77}$ for up- and down-regulated genes, respectively, Fisher's exact test). The median fold changes of up-regulated genes were 1.42 and 1.42, and of down-regulated genes were 1.31 and 1.37, respectively, in the two comparisons. **(C)** Gene ontology analysis for overrepresentation was performed on the genes in common.

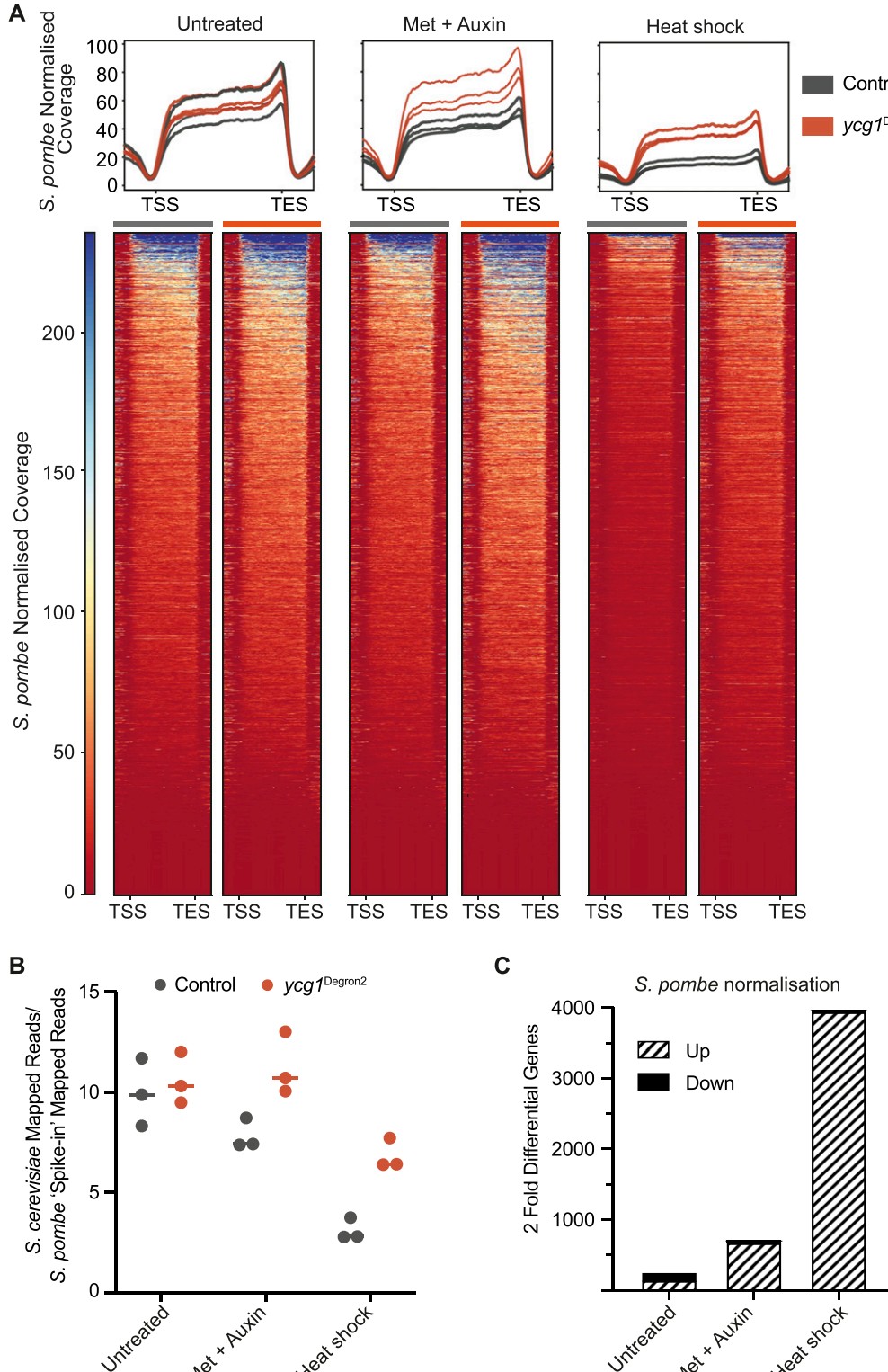

**Figure 6. Condensin promotes global transcriptional shutdown following heat shock.**
**(A)** Metagene profiles and heat maps of transcription of all *Saccharomyces cerevisiae* protein-coding genes, normalised to the *Schizosaccharomyces pombe* spike-in. Compared are control and *ycg1*Degron2 strains untreated, methionine and auxin treated, and after heat shock. Genes are scaled and are flanked by 500 bp up- and downstream of the transcription start site and transcription end site, respectively. The heat map represents the second of the three biological repeats. Metagene profiles summarise the heat maps of all three repeats. The characteristic peak of read coverage close to the transcription end site is likely caused by A-richness surrounding the feature, decoded as "U" during transcription and resulting in 4-TU enrichment (Fig S6B). **(B)** Quantification of overall nascent RNA transcription. The ratio of all mappable nascent *S. cerevisiae* RNA reads relative to the constant *S. pombe* spike-in are plotted for control and *ycg1*Degron2 strains under the indicated conditions. The results from the three biological repeats are shown, the median highlighted. **(C)** The number of greater that twofold differentially expressed genes between control and *ycg1*Degron2 strains, normalised to the *S. pombe* spike-in are shown under the indicated conditions.

global transcriptional repression in human cells exposed to heat shock (Mahat et al, 2016). Global transcriptional down-regulation has also been seen in response to osmotic stress (Rosa-Mercado et al, 2021). When we monitored nascent RNA synthesis by 4-TU

incorporation in response to a variety of different environmental treatments (Fig S6A), we found that transcriptional down-regulation is a common response to environmental changes and stresses in budding yeast.

Two examples of stress response genes are the chaperone *YDJ1* and the Hsp70 family ATPase *SSC1*. Expression of these genes was unaffected by condensin depletion in the absence of stress (Fig 7A). In the control strain, heat shock led to net down-regulation of these genes, even though both genes score as significantly up-regulated after internal normalisation, which equates to the expected transcriptional response to heat shock. In condensin-depleted *ycg1*<sup>Degron2</sup> cells, heat shock barely affected the expression levels of the two stress response genes. This exemplifies how condensin contributes to adjusting gene expression patterns in response to environmental change.

### Condensin and ribosomal protein gene expression

Finally, we returned to analysing the expression of ribosomal protein genes following condensin depletion. We found that expression of the 131 genes encoding components of both the small and large ribosome subunit was elevated in the *ycg1*<sup>Degron2</sup> strain, compared with the control, already before condensin depletion (Figs 7B and S7A). Condensin depletion markedly augmented this difference, resulting in a striking up-regulation of ribosomal protein gene expression. When we then analysed the location of genes whose expression significantly changes in the *ycg1*<sup>Degron2</sup> strain after condensin depletion, we found again a close association with condensin-binding sites. No such association existed in the control strain (Fig S7B). Again, we also saw a striking closeness of genes whose expression changed after condensin depletion to cohesin-binding sites.

The closeness of condensin-binding sites to ribosomal protein genes, whose expression is de-repressed after condensin depletion, is suggestive of a local condensin effect on transcription. On the other hand, ribosomal protein gene expression might be indirectly affected by condensin depletion and the local correlation might be coincidental. Ribosomal protein gene expression is linked to the rates of rRNA synthesis by the Utp22-Ifh1 axis (Laferté et al, 2006; Albert et al, 2016). Consistent with our results obtained by total RNA sequencing, we found that synthesis rates of the large 35S rRNA transcript, *RDN37* produced by RNA polymerase I, increased around 1.5-fold after condensin depletion (Fig S7C). In addition, several genes of the Utp22-Ifh1 axis were amongst those that were significantly more strongly expressed in the *ycg1*<sup>Degron2</sup> strain after condensin depletion, than in the control (Fig S7D). This opens the possibility that increased ribosomal protein gene expression in the absence of condensin is in part a consequence of elevated rRNA synthesis.

In our total RNA sequencing experiment, ribosomal protein gene expression was lower in the *ycg1*<sup>Degron1</sup> strain, compared to the control. While this at first sight contradicts the results obtained with the *ycg1*<sup>Degron2</sup> strain, the two experiments differed in their design. The total RNA sequencing experiment included a shift from minimal to rich growth medium, conditions under which we expect up-regulation of ribosomal protein gene expression. This up-regulation, in turn, could have been compromised in the absence of condensin, thereby resulting in apparently and relatively lower expression in the *ycg1*<sup>Degron1</sup> strain. Together, our observations emphasise the complexity of the mechanisms that adjust ribosomal protein gene expression to cellular needs, which likely involves both direct and indirect contributions from condensin.

## Discussion

We set out in this study to analyse gene expression changes in response to depletion of condensin, a major structural chromosome constituent, in the budding yeast *S. cerevisiae*. Condensin depletion led to widespread gene expression changes. In particular, we realised that condensin plays its most pronounced role in the reprogramming of gene expression patterns in response to environmental stimuli.

When we initially measured steady state transcript levels by mRNA sequencing, 2 h after condensin depletion in G1 arrested cells, around 7% of all genes showed mRNA abundance differences of >1.5-fold compared with the control. The experimental protocol involved a transition from synthetic minimal medium to rich medium and many of the genes that were more highly expressed after condensin depletion fell into the gene ontology "metabolism." In hindsight, knowing about condensin's importance for facilitating gene expression changes, it could be that condensin does not directly regulate metabolic genes, but rather that condensin contributed to dampening their expression in response to the medium change. Ribosomal protein gene transcription, in turn, was lower than in the control, as was the expression of DNA metabolism and cell cycle genes. This could reflect delayed up-regulation of these genes in the absence of condensin, as cells transitioned to a rich nutrient environment.

When we repeated our gene expression analysis, we therefore opted for a smaller environmental change to deplete condensin. The sole noticeable difference to cells was now the addition of methionine to the growth medium (auxin does not have a natural receptor in budding yeast). Using sensitive nascent RNA sequencing to detect gene expression changes, around 11% of all genes showed greater than twofold changes. Notable examples of condensin-regulated genes again included ribosomal protein genes that harbour condensin-binding sites in their promoter (D'Ambrosio et al, 2008).

The most striking contribution of condensin to gene regulation, however, became apparent after heat shock. Here 63% of all genes were differentially expressed in condensin's absence. In almost all cases, these genes partially evaded the widespread transcriptional down-regulation that is inflicted by heat shock. A function in facilitating gene expression changes is consistent with the recently observed role of condensin in executing the transcriptional neuronal maturation programme in flies (Hassan et al, 2020).

How might condensin act in transcriptional regulation? We imagine that condensin acts in more than one way. Condensin is best known for establishing interactions between otherwise distant places on DNA. These could be within the same chromosome or between different chromosomes, for example, between histone gene loci as observed in mammalian cells, or between tRNA loci as seen in yeasts (Haeusler et al, 2008; Iwasaki et al, 2010; Yuen et al, 2017). Such gene clustering could promote transcriptional activity by the formation of transcription hubs where transcription factors and co-factors concentrate. Whether such transcription hubs take the form of phase separated entities (Ryu et al, 2020), or facilitate transcriptional co-regulation in other ways, remains a topic of further study.

In addition to positively co-regulating gene clusters, condensin can act to broadly down-regulate gene expression. This is best seen

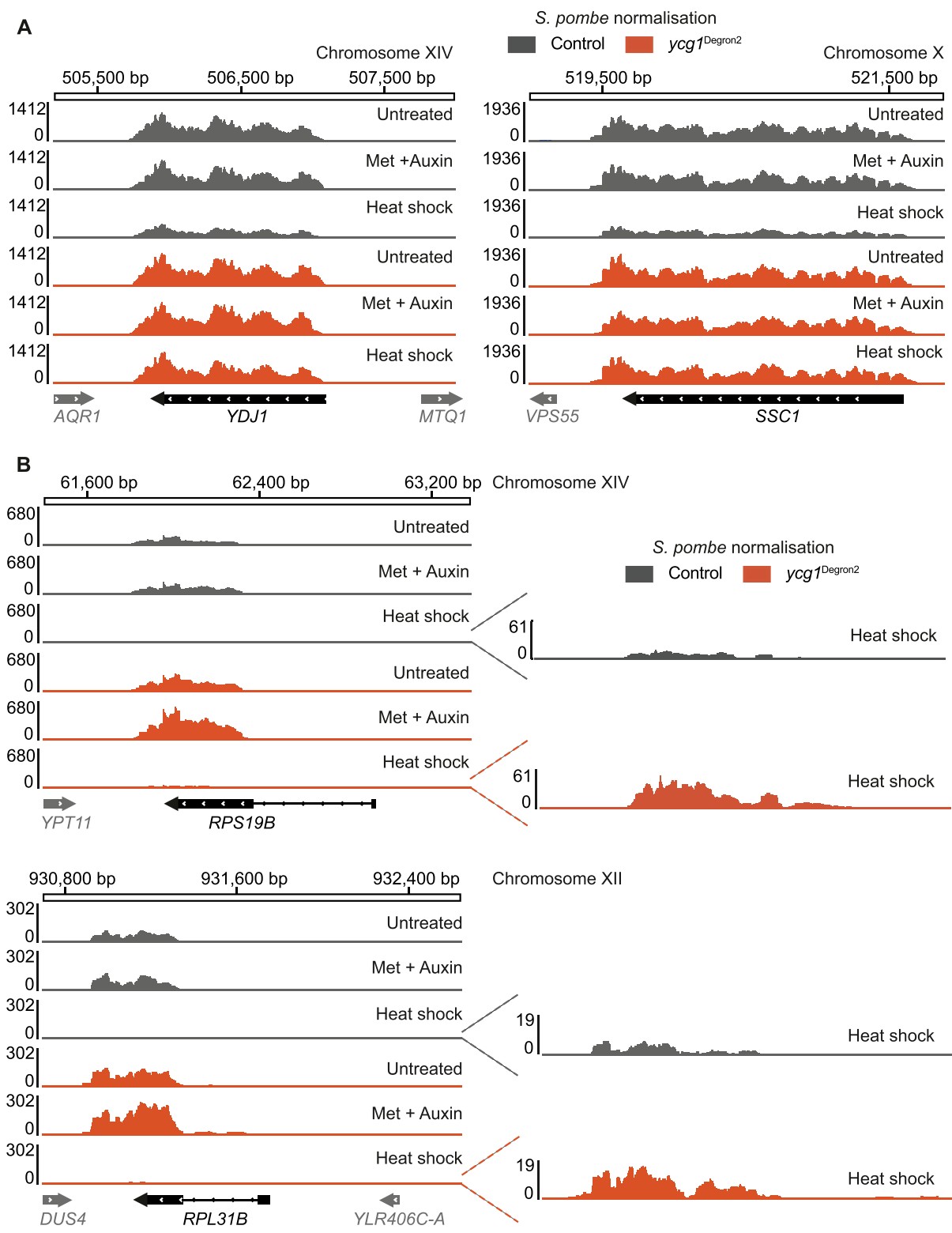

**Figure 7. Impact of condensin on stress response and ribosomal protein gene expression.**
**(A)** Condensin-dependent stress response gene regulation. Nascent RNA synthesis from the *YDJ1* and *SSC1* genes, normalised to *Schizosaccharomyces pombe* spike-ins, in the control and *ycg1*[Degron2] strain under untreated conditions and following methionine and auxin treatment and heat shock. **(A, B)** As (A), but expression of the ribosomal protein genes *RPS19B* and *RPL31B*, components of the small and large ribosomal subunits, is shown.

during transcriptional dosage compensation in the nematode *Caenorhabditis elegans* (Albritton & Ercan, 2018). A condensin complex (condensin[DC]), different from canonical condensin merely by an Smc4 subunit variant, is recruited to the X chromosomes in XX hermaphrodites to halve gene expression from both copies. The mechanism of how condensin[DC] achieves transcriptional down-regulation remains to be fully understood, but it is tempting to speculate that it bears similarities with how budding yeast condensin limits global gene expression in response to heat shock. Condensin[DC] was reported to counteract RNA polymerase II recruitment to promoters (Kruesi et al, 2016). Whether budding yeast condensin similarly prevents RNA polymerase II recruitment to yeast chromosomes will be important to examine. The possible role of histone modifications in conveying condensin's repressive function should be considered (Brejc et al, 2017), as well as condensin's reported ability to counteract DNA unwinding that is central to gene activation (Akai et al, 2011; Sutani et al, 2015).

Yet another mechanism by which condensin impacts on gene regulation could lie in its role to constrain chromatin mobility (Kakui et al, 2017, 2020). Fission yeast condensin has only a marginal impact on the interphase chromatin interaction landscape, but it markedly limits chromatin mobility in the nucleus. Without condensin, chromatin shows the mobility of an unconstrained chromatin chain. By dampening these rapid chromatin movements, condensin might facilitate the establishment of new gene regulatory interactions that define transitions from one transcriptional program to another.

Condensin is an evolutionarily ancient structural chromosome component, around which transcriptional activities have evolved. It is unsurprising that removing this central structural protein complex affects chromosomal processes in multiple ways. In the future, we should explore how transcriptional processes have come to rely on the chromosomal framework provided by condensin and how certain gene expression processes have harnessed condensin's potential and put it to their use.

# Materials and Methods

## Cell culture and cell synchronisation

Budding yeast cells were grown in minimal YNB medium (Yeast Nitrogen Base, 0.8% wt/vol) supplemented with CSM-methionine (Complete Supplement Mixture; Formedium) and 2% glucose wt/vol. Mid-log phase cells of the a mating type were arrested for 1.5–2 h after the addition of $\alpha$-factor (WHWLQLKPPGQPMY, provided by the Crick Peptide Chemistry Science Technology Platform, used at a final concentration of 7.5 $\mu$g/ml from a 5 mg/ml stock in methanol). Pheromone was added again after 1 h. For release, cells were filtered and washed in at least three culture volumes of YNB or YP (yeast peptone; 1.1% wt/vol yeast extract, 2.2% wt/vol bacto-peptone, and 0.0055% wt/vol adenine) medium, depending on the desired release medium. Cells were then resuspended in fresh YNB + CSM medium (with or without 133 $\mu$g/ml methionine added) or Yeast Peptone medium, both supplemented with 2% (wt/vol) glucose. Auxin (indole-3-acetic acid; Sigma-Aldrich) was added at a

final concentration of 88 $\mu$g/ml. $\alpha$-factor was re-added after cells had budded after around 45 min for re-arrest in the following G1 phase. Alternatively, nocodazole (10 $\mu$g/ml final; Sigma-Aldrich) was contained in the release medium to impose a mitotic arrest. For the spot dilution assay, mid-log phase cultures were all adjusted to $OD_{600}$ = 0.2. Drops of serial fivefold dilutions were the applied onto plates containing 2% glucose (wt/vol) and either YNB + CSM-methionine or YPD + Auxin (88 $\mu$g/ml). Cells were grown at 25°C for 3–5 d. Genotype details of the budding yeast strains used in this study, as well as of plasmids used for their construction, can be found in Table S2.

## Immunoblotting

Cell aliquots were fixed in trichloroacetic acid, and cell extracts were prepared and separated by SDS–polyacrylamide gel electrophoresis, before their transfer to a nitrocellulose membrane. Antibodies used for detection are listed in Table S2. Visualisation was through enhanced chemoluminescence (ECL) reagents (GE Healthcare).

## Immunofluorescence

Aliquots of the culture were fixed overnight in cold fixation buffer (100 mM potassium phosphate, pH 6.4, 0.5 mM $MgCl_2$, and 3.7% formaldehyde). Cells were spheroplasted in a buffer containing 28 mM $\beta$-mercaptoethanol and 20 U/ml Zymolase T-100 (Amsbio) by incubation at 37°C for 45 min. Immunofluorescence staining was performed using antibodies against Nop1 to evaluate nucleolar segregation and against $\alpha$-tubulin to visualise progression through mitosis. Cells were also stained with the DNA dye 4′,6-diamidino-2-phyeylidindole (DAPI) before mounting with ProLong Gold antifade (Invitrogen). Fluorescent images were acquired using a DeltaVision imaging system (Applied Precision) based on an Olympus IX-71 microscope.

## FACS analysis of DNA content

Cells were fixed for 1 h in cold 70% ethanol and treated with 0.1 mg/ml RNase A for 4 h. DNA was stained with 50 $\mu$g/ml propidium iodide in FACS buffer (200 mM Tris–HCl, pH 7.5, 211 mM NaCl, and 78 mM $MgCl_2$). The samples were sonicated and run on a FACSCalibur Cell Analyzer (BD Biosciences) using CellQuest software, before further analysis in FlowJo v.10.

## Chromosome size analysis by pulsed-field gel electrophoresis

Chromosomes were resolved and rDNA detected as described (El Hage & Housley, 2013), with the following alterations. Cells were collected from asynchronously growing cultures in mid-log phase without condensin depletion. To resolve *S. cerevisiae* chromosomes, the gel was run with a 300–900 s switch time, 120° angle, 3 V/cm at 14°C for 68 h. The gel was stained with GelRed (Biotium) in TBE for 1 h, washed twice in 2× TBE for 15 min, and imaged. The gel was transferred onto an N+ Hybond nitrocellulose membrane (GE Healthcare) through capillary transfer as described (El Hage & Housley, 2013). The rDNA probe for Southern blotting was

amplified from the non-transcribed spacer region two (NTS2). The probe was labelled with [$\alpha$-$^{33}$P] dATP (3,000 Ci/mmol; Hartmann) using the Prime-It II Random Primer Labelling Kit (Agilent). The probe was then added to the membrane pre-hybridised in QuickHyb Hybridization solution (Agilent), and incubated at 68°C overnight. Membranes were washed twice in 2× SSC, 0.1% SDS for 15 min at room temperature, and twice in 0.5× SSC, 0.1% SDS, rinsed in 50 mM Tris–HCl, pH 7.5 and exposed overnight using a Phosphor-Imager screen (Amersham Biosciences), and scanned on a Typhoon 9400 Imager.

### RNA extraction

Yeast cells were collected and washed in ice-cold water before being snap frozen in liquid nitrogen. The RNA was isolated using acid phenol:chloroform:isomyl alcohol (125:24:1; Ambion), and precipitated in ethanol containing 0.3 M sodium acetate. RNA was reconstituted in DEPC-treated water and purified using the RNA Cleanup protocol within the RNeasy Mini Kit including DNase digestion (QIAGEN). RNA integrity was assessed on a 2100 Bioanalyzer (Agilent), and the RNA concentration determined on a Qubit Fluorometer using either the Qubit RNA Broad-Range or High-Sensitivity Assay Kit (Thermo Fisher Scientific).

### cDNA synthesis and quantitative real-time PCR (qPCR)

cDNA was synthesised from 2 μg of total RNA using SuperScript III Reverse Transcriptase (Thermo Fisher Scientific). PowerUp SYBR Green Master Mix (Thermo Fisher Scientific) was used for all qPCR reactions. A dilution series of commercial S. cerevisiae DNA (Novagen) was used to ensure linear amplification. Control genes for normalisation were ACT1 or UBC6, the levels of which were monitored between conditions relative to the S. cerevisiae DNA dilution series and were found to remain constant. Reactions were set up in 384 well plates that were then run on a QuantStudio 12, QuantStudio 7K Flex Real-Time PCR System, or a QuantStudio 5 machine. The qPCR run involved a hold at 50°C for 2 min, and 95°C for 10 min followed by 50 cycles of 95°C for 15 s and 60°C for 1 min. Primer pairs used for amplification are listed in Table S2.

### Total and mRNA library preparation

RNA was extracted as described above. mRNA libraries were made using the KAPA mRNA HyperPrep kit (Roche), whereas total RNA libraries were made using the KAPA RNA HyperPrep kit without RiboErase treatment to include ribosomal RNAs.

### Nascent RNA isolation for TTChem-Seq

TTchem sequencing was developed for use with human cells (Schwalb et al, 2016; Gregersen et al, 2020). This protocol was adapted for use with Saccharomyces cerevisiae and involves the use of 4-thiouracil (4-TU), as opposed to 4-thiouridine (4-SU). Cells were pulsed for 5 min through addition of 5 mM 4-TU to the growth medium from a 1M stock solution in DMSO. After RNA extraction, 90 μg of each S. cerevisiae RNA sample was combined with 10 μg from a 4-TU pulsed S. pombe RNA stock. Fragmentation time was increased

to 30 min compared with the published protocol. The streptavidin beads for isolating the nascent biotinylated RNA were washed twice at 55°C with 500 μl pull out wash buffer (100 mM Tris–HCl, pH 7.4, 10 mM EDTA, 1 M NaCl, and 0.1% Tween-20), three times with 500 μl TE buffer (10 mM Tris–HCl, pH 7.4, and 1 mM EDTA), and twice at room temperature with 500 μl EB buffer (10 mM Tris–HCl, pH 8.5). The samples were eluted in 100 μl of 10 mM DTT in EB buffer at room temperature with a second elution 5 min later.

### Dot blot to visualise 4-TU incorporation

An aliquot of 4-TU pulsed total RNA was taken after biotinylation. The sample concentration was adjusted to 1 μg/μl. 1 μl of each adjusted sample was then spotted onto a Hybond-N Membrane (GE healthcare) and the RNA was crosslinked onto the membrane using a Stratalinker 1800 UV (120,000 μJ). The membrane was blocked for 20 min at room temperature in blocking buffer (10% SDS in PBS and 1 mM EDTA), and probed with a 1:50,000 dilution of HRP-conjugated streptavidin for 15 min at room temperature. The membrane was then washed twice for 10 min in each of the following buffers: blocking buffer, wash buffer I (1% SDS in PBS), and wash buffer II (0.1% SDS in PBS). The biotin bound RNA was visualised using Amersham ECL Western Blotting Detection Reagents (GE Healthcare) and an Amersham Imager 600 (GE Healthcare).

### Nascent RNA library preparation

20 ng of nascent RNA was used to make libraries with the KAPA RNA HyperPrep Kit (Roche). The library was made following the manufacturer's instructions, with adjustments to ensure small transcripts from the fragmentation were not lost. The ratio of KAPA pure beads to adapter-ligated cDNA was 0.95× in the first post-ligation cleanup, and 1× in the second post-ligation cleanup. The cDNA library was amplified with 11 cycles of PCR.

### Genome-wide RNA-Seq analysis

RNA sequencing was carried out on the Illumina HiSeq 4000 platform and typically generated ~18 million 76 bp strand-specific single-end reads per sample. Adapter trimming was performed with cutadapt (version 1.9.1) (Martin, 2011) with parameters "--minimum-length = 25 –quality-cutoff = 20 -a AGATCGGAAGAGC." The RSEM package (version 1.3.0) (Li & Dewey, 2011) in conjunction with the STAR alignment algorithm (version 2.5.2a) (Dobin et al, 2013) was used for mapping and subsequent gene-level counting of the sequenced reads with respect to all S. cerevisiae genes downloaded from the Ensembl genome browser (assembly R64-1-1, release 90). ERCC spike-in sequences were appended to the genome fasta and GTF annotation for quantification. The parameters used were "--star-output-genome-bam–star-gzipped-read-file–forward-prob 0," and all other parameters were kept as default. Mapping statistics from both sequencing experiments can be found in Table S3.

Differential expression analysis was performed with the DESeq2 package (version 1.12.3) (Love et al, 2014) within the R programming environment. A false discovery rate of 0.05 or less was used as the significance threshold for the identification of differentially expressed genes. The counts corresponding to total and polyA were

normalised individually and together, omitting genes with names ending in "rRNA" or "RDN" as were ERCC spike-ins. The design was as follows: "phase × treatment" for the individually normalised data to allow estimation of the treatment effect in both phases; or "batch + phase × treatment" for combined total and polyA, where *batch* allows for a systematic difference in baseline between total and polyA.

To perform the principal component analysis, the normalized counts were variance-stabilized using the rlog function within DESeq2. The samples were projected onto the principal components using the standard R function princomp on all genes without further scaling.

The overlap of genes shown in Venn diagrams was assessed through a Fisher's Exact test which was performed on R using the GeneOverlap package, based on 6,049 genes within the *S. cerevisiae* genome.

To assess the proximity of differential genes to genome features of interest, we calculated the distance from each transcript to its nearest neighbour in the feature-set. These distances were bi-partitioned by whether the transcript be significant or null, and the ratio of the median distance in each partition was calculated (in cases where those medians were zero, due to most genes over-lapping a member of the feature-set, we replaced the median by the tally of how many were zero-distance). This was repeated for each differential list and each feature-set. For example, the median distance of significantly changed genes in the G1 total RNA se-quencing dataset from their nearest condensin-binding site was 3,916 bp, whereas the median distance of unchanged genes was 8,382 bp. These distances do not by themselves carry biological meaning, but dividing one by the other leads to the reported proximity association value of 0.47.

### Genome-wide nascent RNA-Seq analysis

Nascent RNA sequencing was carried out on the Illumina HiSeq 4000 platform and typically generated ~19 million 76 bp strand-specific single-end reads. Adapter trimming was performed with cutadapt (version 1.9.1) (Martin, 2011) with parameters "--minimum-length = 20 –quality-cutoff = 20 -a AGATCGGAAGAGC." BWA (version 0.5.9-r16) (Li et al, 2009) using default parameters was used to perform the read mapping independently to both the *S. cerevisiae* (assembly R64-1-1, release 90) and *S. pombe* (assembly ASM294v2, release 44) genomes. Genomic alignments were filtered to only include those that were primary, uniquely mapped, and had fewer than three mismatches using BamTools (version 2.4.0 [Barnett et al, 2011]). Alignments corresponding to the sense and antisense strands were obtained using SAMtools view (version 1.3.1) (Li et al, 2009) by using the flags "-f 16" and "-F 20," respectively. Read counts relative to protein-coding genes were obtained using the featur-eCounts tool from the Subread package (version 1.5.1) (Liao et al, 2014). The parameters used were "-O -s 2."

BedGraph tracks were created using the BEDTools genomeCo-verageBed (version 2.26.0) (Quinlan & Hall, 2010) by normalising the genome-wide coverage relative to DESeq2 size factors generated with respect to the *S. pombe* transcriptome. The parameters used were "-bg -pc -strand <STRAND> -scale <SCALE_FACTOR>." BedGraph files were converted to bigWig using the wigToBigWig binary available from the UCSC with the "-clip" parameter (Kent et al, 2010).

The computeMatrix scale-regions command from the deepTools package (version 2.5.3) (Ramírez et al, 2016) was used to generate coverage matrices with respect to *S. cerevisiae* protein-coding genes. The parameters used were "--regionBodyLength 1,000 –beforeRegionStartLength 500 –afterRegionStartLength 500 –binSize 10 –missingDataAsZero–sortRegions no–scale 1." Meta-profile plots were generated directly from the output of computeMatrix with the plotHeatmap command from the deepTools package.

Two different normalisation schemes were applied: an internal normalisation where only *S. cerevisiae* transcripts contributed to the calculation and a cross-normalisation where, conversely, we only used the spiked-in *S. pombe* transcripts to assess the size factors, potentially mitigating any impact a global shift in the *S. cerevisiae* read counts would have on DESeq2's usual normalisation procedure. Other than that, a standard DESeq2 analysis was carried out, in both cases using a "strain × treatment" to allow contrasts to be drawn between strains within a treatment group, and also between treatments within a strain. Heat maps were generated on the variance stabilized ("vst" from DESeq2) normalized counts using complete linkage of Euclidean distances between samples and transcripts, scaled so that the average control treatment was zeroed for the colour scheme.

### Repetitive read analysis

Adapter trimming was performed with cutadapt (version 1.9.1) (Martin, 2011) as described in the previous sections. BWA (version 0.5.9-r16) (Li & Durbin, 2009) was used to perform the read mapping to the *S. cerevisiae* genome. To obtain all possible multi-mapping read locations, "-R 1000000" and "-n 1000000" were specified when running "bwa aln" and "bwa saitosam," respectively. Reads were filtered to only include those that had fewer than three mismatches using BamTools (version 2.4.0 [Barnett et al, 2011]); Custom scripts written in Python using the Pysam package (version 0.9.0; https://github.com/pysam-developers/pysam) were then used to create an "expanded" alignment file whereby the best possible mapping lo-cations for each read were appended as distinct entries. Additional scripts were written to count the reads relative to transposon in-tervals while normalising for the total number of mapped reads per million and by weighting for the total number of mapping locations.

### Gene ontology analysis

Statistical overrepresentation analysis of greater than twofold differentially expressed genes compared to the *S. cerevisiae* ge-nome was conducted on Panther using GO-Slim Biological process (Thomas et al, 2006). Where terms were closely related, the term with greatest statistical overrepresentation was selected.

## Data Availability

The gene expression data from this publication have been de-posited to the gene expression omnibus database and assigned the identifier GSE161582 (https://www.ncbi.nlm.nih.gov/geo/query/acc.cgi?acc=GSE161582).

## Supplementary Information

## Acknowledgements

We would like to thank F Van Wervern and his lab for guidance on RNA extraction, L Gregersen and J Svejstrup for the introduction to TT$_{chem}$-seq, the Crick Advanced Sequencing Science Technology Platform for high throughput sequencing, J Houseley for the *NTS2* probe, A Morillon for transposon primer sequences, C Bouchoux for her help and all our lab-members for discussion and comments on the manuscript. This project received funding from the European Research Council (ERC) under the Horizon 2020 program (grant agreement No. 670412) and from The Francis Crick Institute, which receives its core funding from Cancer Research UK (FC001198), the UK Medical Research Council (FC001198), and the Wellcome Trust (FC001198).

### Author Contributions

L Lancaster: conceptualization, data curation, formal analysis, investigation, and writing—original draft.
H Patel: data curation, formal analysis, and visualization.
G Kelly: data curation, formal analysis, and visualization.
F Uhlmann: conceptualization, formal analysis, supervision, funding acquisition, project administration, and writing—review and editing.

### Conflict of Interest Statement

The authors declare that they have no conflict of interest.

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
