## [Reviewer comments · Life Science Alliance]

Life Science Alliance

A role for condensin in mediating transcriptional adaptation to environmental stimuli

Lucy Lancaster, Harshil Patel, Gavin Kelly, and Frank Uhlmann

DOI: <https://doi.org/10.26508/lsa.202000961>

Corresponding author(s): Frank Uhlmann, The Francis Crick Institute

Review Timeline:	Submission Date:	2020-11-17
	Editorial Decision:	2021-01-07
	Revision Received:	2021-03-29
	Editorial Decision:	2021-05-19
	Revision Received:	2021-05-21
	Accepted:	2021-05-21

Scientific Editor: Shachi Bhatt

Transaction Report:

January 7, 2021

Re: Life Science Alliance manuscript #LSA-2020-00961

Dr. Frank Uhlmann
The Francis Crick Institute
Chromosome Segregation Laboratory
1 Midland Road
London NW1 1AT
United Kingdom

Dear Dr. Uhlmann,

Thank you for submitting your manuscript entitled "A role for condensin in mediating transcriptional adaptation to environmental stimuli" to Life Science Alliance. The manuscript was assessed by expert reviewers, whose comments are appended to this letter.

As you will note from the reviewers' comments, the reviewers are quite intrigued by these findings but have raised a number of important constructive concerns that should be addressed. We would thus like to invite you to submit a revised manuscript that addresses all of the reviewers concerns.

Thank you for this interesting contribution to Life Science Alliance. We are looking forward to receiving your revised manuscript.

Sincerely,

Shachi Bhatt, Ph.D.

Executive Editor
Life Science Alliance
<https://www.lsjournal.org/>
Tweet @SciBhatt @LSAJournal

- A letter addressing the reviewers' comments point by point.
- An editable version of the final text (.DOC or .DOCX) is needed for copyediting (no PDFs).
- High-resolution figure, supplementary figure and video files uploaded as individual files: See our detailed guidelines for preparing your production-ready images, <https://www.life-science-alliance.org/authors>
- Summary blurb (enter in submission system): A short text summarizing in a single sentence the study (max. 200 characters including spaces). This text is used in conjunction with the titles of papers, hence should be informative and complementary to the title and running title. It should describe the context and significance of the findings for a general readership; it should be written in the present tense and refer to the work in the third person. Author names should not be mentioned.

B. MANUSCRIPT ORGANIZATION AND FORMATTING:

Reviewer #1 (Comments to the Authors (Required)):

In this manuscript, the authors addressed how depletion of condensin, a key player in chromosome organization, affects gene regulation in budding yeast. They observed relatively small, but reproducible change in transcription level across the genome. In particular, condensin removal attenuated heat shock-dependent global transcription repression.

The important point of this work is use of AID system, which allows ligand-dependent rapid degradation of a target protein. This enabled the authors to detect transcriptional change caused solely by condensin depletion. To my knowledge, it is the most carefully designed study to

investigate condensin's role in transcription regulation so far. Though it remains to be elucidated how condensin functions in transcription control, this work provides a valuable insight to the researchers in chromosome biology field.

For the most part, the experiments were rigorously controlled and data analysis was properly conducted. I agree that the main points of this manuscript are supported by the data sufficiently, except a major concern shown below.

I recommend this paper for publication in *Life Science Alliance*, once this major concern as well as the minor issues listed below are fully addressed.

Major concern.

Page 14 line 23 and Fig 7B: The authors mentioned that stress response genes were down-regulated in wt cells treated by heat shock. This seems against what we know about the expression of heat shock genes. I wonder if heat shock affects uptake or metabolic conversion of thioracil, thereby reducing thiol-labeled "nascent RNA" signal. If this is the case, normalization based on the spike-in control did not work properly for the heat shock condition and (at least some of) the conclusions drawn from the nascent RNA-seq analysis are inappropriate. I would like the authors to validate the result of nascent RNA-seq (Figs. 6 and 7) by another experimental method and resolve the above concern. Calibrated ChIP-seq of RNAPol2 may be a possible approach.

Minor points.

- Page 7 line 25 and Fig 1C: describe which condensin ChIP-seq data was used. Provide the calculated mean distance between the differentially expressed genes and the nearest condensin binding sites (along with that for control).
- Page 12 line 5 and Fig 5B: Provide lists of the detected differentially expressed genes. These are valuable information to be shared.
- Read mapping: provide mapping statistics (i.e., total read number, number of mapped reads and its proportion) in a supplementary table.
- Principal component analysis: explain how the analysis was conducted.
- t-test: Please double-check the calculated p-values are correct, particularly for Fig 2A. I feel p-value of 0.0063 is too good for such large SD and small sample size.
- Page 17 line 24: it seems fair to cite also Dr Shirahige group's work (PMID: 26204128) for condensin's role in DNA rewinding.

Reviewer #2 (Comments to the Authors (Required)):

LSA - Review Lancaster et al.

In this manuscript, Lancaster et al. investigate whether condensin plays a role in the control of gene expression using budding yeast as a model system. Condensin is best characterized as the key driver of mitotic chromosome assembly or condensation. Whether or not condensin takes part in the control of gene expression remains an open and controversial question. Lancaster et al. tackled this problem by analysing the transcriptome upon inactivation of condensin using degron alleles, in budding yeast cells arrested in either G1 or early - M (i.e. prior to anaphase). This experimental setting is perfectly appropriate to avoid the possible confounding effect of chromosome instability on gene expression, caused by lack of condensin in anaphase. The authors observed changes in the transcriptome in G1 or early-M, amongst which differential expression of ribosomal protein coding genes (RPGs). Since RPGs are bound by condensin, the authors suggest that condensin

might control gene expression in-cis. Based on nascent RNA-seq data, they further suggest that condensin plays a role in transcriptional downregulation in response to heat shock (environmental stress).

These results are potentially important given the current controversy regarding the role played or not by condensin in the control of gene expression. However, I see several potential weaknesses in the current version of the manuscript that challenge the conclusions raised by the authors.

Amongst the most salient ones are (1) the functionality of the degron alleles of condensin used in this study, in ON condition, and the possibility that phenotypes observed after auxin-mediated depletion might in fact stem from, or be primed by, pre-existing defects during asynchronous growth, and (2) the possibility that changes in RPGs expression level might be the indirect consequence of changes in RNA Pol I transcription, as explained below.

There is evidence in the manuscript that the degron alleles of condensin used by the authors are not fully functional in ON growth condition. The steady state level of Ycg1-degron1 is strongly reduced, even in the absence of auxin, and the authors have shown the cognate strain exhibits a ~50% reduction in the number of rDNA repeats, indicating that condensin is impaired prior to auxin-mediated depletion, hence raising the possibility of confounding effects prior to cell cycle arrest and auxin-mediated depletion. Similarly, the ygc1-degron2 strain exhibits increased RPG expression prior to auxin adjunction (Fig. 7). Condensin is abundant in the nucleolus and play a key role in the structural integrity of rDNA repeats. Transcription of RPGs by Pol II is tightly coupled to RNA Pol I activity, which is itself tightly linked to the number of rDNA repeats. Thus, instead of a direct local control of RPG transcription by condensin, there is a possibility that changes in RPG expression observed upon condensin inactivation might be the indirect consequence of a chain reaction initiated at the level of the rDNA and propagated through an axis that connects the level of rRNA to transcription of RPGs. Although I acknowledge the importance of the work and its potential impact in the field, I suggest that several important control experiments must be performed in order to ascertain the validity of the conclusions raised by the authors.

Major points

(1) The functionality of the degron alleles prior to auxin-mediated depletion.

The authors used two different degron versions of condensin in the course of their work: ycg1-degron 1 in Fig. 1-4, and ygc1-degron2 in Fig. 5-7.

In Fig. 1, 4 and S2B, the authors show that the steady state level of the Ycg1-degron 1 protein is reduced compared to control, even in ON condition (no auxin), and that the strain exhibits a ~ 50% reduction in rDNA copy number, indicating that the YCG1-degron 1 allele is impaired even in the absence of auxin. However, the consequence of this constitutive impairment on chromosome segregation and RNA level is not documented.

To assess the impact of condensin on gene expression, the authors compared the transcriptome, by using RNA-seq, between a control and the YCG1-degron 1 strain after auxin adjunction, (Fig. 1), but not prior to auxin adjunction. In that context, it is not possible to determine the impact on RNA levels (1) of the degron tag itself, and (2) of the constitutively reduced level of Ycg1 prior to auxin adjunction. Thus, the origin of the observed differences remains ambiguous. Since, as the authors mentioned, Ycg1 is a limiting parameter for condensin function, changes in RNA level detected in the ycg1-degron1 strain might stem, at least in part, from the constitutively limited condensin activity prior to G1 arrest.

I suggest that the authors should compare the transcriptome of a Ygc1-degron1 strain in ON vs OFF conditions, ideally by RNA-seq, and at minima by comparing various representative reporter genes by RT-qPCR, notably ribosomal protein coding genes (RPGs). Also, the state of chromosome segregation, notably of the nucleolus, should be examined in both the ygc1-degron1 and -degron2 strains in ON vs OFF conditions to assess whether aneuploid cells might pre-exist to the cell-cycle

arrests and auxin adjunction.

(2) Spatial correlation between gene deregulation and condensin localisation

In Fig. 1C, the authors describe a positive spatial correlation between condensin binding sites and RPGs differentially expressed in G1, which could suggest that condensin controls gene expression in cis. However, the spatial correlation is obvious when they interrogate small-size Total RNA samples, but far less when they interrogate larger mRNA samples. I find difficult to understand this difference since RPG mRNA should be present in the mRNA samples, unless the spatial correlation results from the strong difference in the size of the interrogated samples, i.e. the statistical significance obtained when analysing the small size Total RNA samples (n=356 and 203) becomes weaker with larger mRNA samples (n=1110 and 868). Could this point be made clearer? This is especially important because analyses conducted on larger mRNA samples suggest that a large part of deregulated mRNAs does not lie in the vicinity of condensin binding sites, and, thus, that condensin might not control all differentially expressed genes at the local level.

(i) Whether ribosomal protein coding mRNA are up or down-regulated is not indicated, (ii) the amplitude of the change is also not indicated (is it 1.5 or more?), and (iii) given the potential importance of this finding, it would be important to confirm the changes in RPG expression by RT-qPCR.

Also, though the vast majority of differentially expressed RNAs identified in Total RNA samples overlap with those identified by mRNA sequencing, the reverse is not true. More than 70% of mRNA identified as differentially expressed are absent for Total RNA samples. Thus, it remains unclear which type of sample is the most relevant to assess the impact of condensin on gene regulation.

What is the statistical significance of the overlap shown in Fig. S1 by hypergeometric test?

In the same vein, a positive spatial correlation is detected with cohesin binding sites (Fig. 1C), but it remains unclear whether this is related to the presence of condensin at sites also occupied by cohesin, or instead the indirect consequence of the down regulation of genes involved in sister-chromatid cohesion (Fig. 1B). Please clarify.

(3) rDNA transcription and RPG expression

The authors assessed rRNA expression by quantifying the short-lived ITS1 (spacer) transcripts, and observed no change in G1 and a small increase during M phase by RT-qPCR. ITS1 expression levels have been normalised to the Pol 2 transcribed UBC6 gene (which should be shown as not differentially expressed). However, since the rDNA copy number is reduced by ~50% in the ycg1-deg1 strain compared to the control (Fig. S2), unchanged ITS1 RNA levels imply that transcription of the few remaining rDNA units is increased, a change that could not be detected through a normalization with UBC6. There is robust evidence that reduced rDNA copy number is compensated by an increased expression level of the remaining copies (Ide et al. Science 2010). I suggest that authors should normalise their measurements of ITS1 RNA levels to the total rDNA copy-number (determined by qPCR) present in the strains.

Importantly, if a change in expression of rRNA is detected, I suggest that the authors should further assess the possibility of a causal relationship with the differential expression of RPGs. Indeed, the transcriptional activity of RNA polymerase I is a key determinant for the expression level of RPGs, notably through a molecular titration mechanism involving the rRNA binding protein Utp22 and the RPG transcription factor Ifh1 (Laferte, A. et al. Genes Dev. 2006, 20, 2030-2040; Albert, B. et al. Mol. Cell 2016, 64 (4), 720-733). This is crucial to ascertain the origin (direct or indirect) of differential RPG expression in condensin-depleted cells.

(4) Transposon activation

The authors suggest that Ty2 transposon expression is up-regulated upon stress, and that condensin-facilitates this up-regulation. However, looking carefully at the data shown Fig. 4, I see

almost no increase in Ty2 expression level in control cells experiencing stress (panel B) versus no stress (panel C): 0.15 vs 0.15; 0.25 vs 0.2 and 0.3 vs 0.2. Please clarify.

Also, the steady state level of act1 used to normalize the data should be provided in both conditions.

(5) Nascent RNA analyses using YCG1-degron2 strain.

This part of the manuscript is clearly the most robust from an experimental point of view. The authors used a different ycg1-degron 2 allele that showed greater stability than ycg1-degron1 prior to auxin adjunction, and measured the level of nascent RNA. To take into account the impact of the degron2 tag, the authors first compared ycg1-degron2 in ON vs OFF condition, which identified 772 genes. They also compared the control and Ycg1-degron 2 strains, both treated with auxin, which should eliminate the impact of auxin itself. Finally, they intersected the two ensembles and identified 506 genes up-regulated and 236 genes down-regulated, by more than two-fold.

However, the authors provide no further information on (i) the amplitude of the changes they observed, (ii) gene ontology, and, importantly, (iii) the spatial correlation with condensin binding sites, upon auxin adjunction, and upon heat-shock. These data should be provided. A list of genes common to fig.1 and fig. 4 should also be provided. These analyses should prove extremely valuable given that the results are likely to be more robust than those shown in Fig. 1.

With respect to the comparison between the control and the ycg1-degron2 strain prior to auxin adjunction, a difference in gene expression is manifest in Fig. S5A and in Fig. 7, and mentioned by the authors in the text (page 12). Yet, at page 113, it is stated that nascent transcription is comparable in the two strain, based on metagene profiles. Please clarify.

On the impact of heat shock on gene expression in control vs ycg1-degron2 strains (Fig. 6). Looking at the data in Fig. 6, it seems to me that nascent transcription drops in the control strain upon auxin adjunction, whilst in the ycg1-degron2 strain the normalized coverage remains either constant (Fig. 6B) or slightly increases (Fig. 6A). Importantly, it also seems to me that the average coverage drops upon heat shock with a similar amplitude in the two strains; the initial delta prior to heat shock remaining roughly constant after heat shock. A similar behaviour is visible also for RPGs in Fig 7. If this is correct, it would suggest that condensin is required for a response to a possible stress caused by auxin, rather than for the down regulation of genes upon heat shock. Could you please make this point clearer.

Also, I find difficult to reconcile the data shown in Fig. 6C with the heat maps. Looking at the heat maps it seems to me that more than 750 genes are up-regulated in ycg1-degron2 strain in Met+auxin condition compared to the control. Could you please verify.

(6) The nascent transcription of highly expressed genes.

Looking at the data in Fig. 7A, I see a clear 2-fold increase in the transcription level of the RPS19B and RPL31B ribosomal protein genes in the ycg1-degron2 strain compared to control even in untreated condition (i.e. prior to auxin adjunction). This substantiates the idea of a pre-existing defect, which might have primed the further increase seen upon auxin adjunction. This point deserves a more detailed description. I suggest that the authors provide the full distribution of fold changes observed at RPGs between the control and Ycg1-degron 2 strain grown in the untreated condition, and in the treated condition, to allow for a comprehensive comparison.

Furthermore, related to the link between RPGs and rDNA transcription, and to rule out any indirect effect, the authors should measure rRNA (ITS1) levels in the ycg1-degron2 strain prior and after auxin-adjunction, taking into consideration the number of rDNA repeats in the strain, and also verify the expression level of the factors involved in the Utp22/Ish1 axis that connects rRNA level to RPG expression.

(7) tRNA gene expression.

Condensin is bound to tRNA genes in budding yeast. According to the idea that condensin controls gene expression at the local level in budding yeast, one would expect to observe changes in tRNA levels upon condensin depletion. Could the author perform this important verification?

Minor points

- Introduction, page 3. It is mentioned that budding yeast condensin has been found at promoters of highly expressed genes (other than tRNA genes), but I am unaware of any published data supporting this. Please clarify.
- Fig. 1B vs Fig. S1. The number of differentially expressed genes do not correspond. Ca. 450 genes in G1 in Fig. 1B vs 1100 in Fig. S1. Please clarify.
- Figure S2. Legend mentions FIG1 mRNA, but images show AGA2
- Fig. S5A. Please, check for typo *ycg1-degron1* vs *-degron2*.
- Statistical significance of overlaps calculated by hypergeometric test should be indicated in Fig. 5B, Fig. S5B-C.
- Finally, could the authors discuss why in their opinion a difference in gene expression upon heat shock has not been detected by Hocquet et al. 2018 or Nakazawa et al. 2015 in fission yeast, upon inactivation of condensin using *ts* alleles. Could it be linked to the different spatiotemporal dynamics of condensin in budding and fission yeasts?

Reviewer #3 (Comments to the Authors (Required)):

Condensin is a major organiser of chromosome architecture best known for its role in promoting DNA compaction during mitosis. Condensin also binds DNA during interphase and was reported to influence gene transcription in several organisms, cell types and experimental conditions. Yet no clear picture has emerged. In the present study, the authors developed an experimental system aiming at measuring transcription changes upon rapid depletion of condensin in staged budding yeast cells. Their main claim is that condensin is required for gene expression changes in response to environmental cues. This idea emerged during the course of the study. Their experimental system to degrade condensin required a change in the culture medium, from poor to rich. In G1 cells condensin depletion led to small expression changes in about 7% of genes. They observed that a particular class of transposon was down-regulated. Since transposon mobilisation may provide a selective advantage upon environmental changes they hypothesized that condensin may have a role in this process. Indeed, they found transposon expression changes were dependent on the sudden modification of the culture medium and facilitated by condensin. Following this idea, the authors moved to the response to heat-shock (HS) and present data arguing that the general transcriptional shut down in response to HS was largely dependent on condensin.

A role for condensin in facilitating transcriptional adaptation to environmental cues is appealing and would be of great interest to a large audience. I have however concerns about the experimental system that cast doubts about the main conclusions drawn by the authors, as detailed below.

Main points

The experimental setup aimed at rapidly depleting condensin in staged cells to avoid confounding variables. To do so, the *YCG1* gene was tagged with a degron and a PK epitope for detection. In addition the *YCG1* promoter was replaced by the *MET3* promoter to shutoff expression by the addition of methionine in the culture medium. The resulting strain showed an altered gene expression profile and a reduced number of rDNA repeats. The strain clearly does not behave as

wild-type meaning that the initial conditions (before condensin depletion) are far from normal. It was further complicated as the authors switched to another version of the degron strain during the course of the study. I found difficult to extract the effects attributable to condensin depletion from pre-existing defects of the degron strains. I suggest the authors include some additional controls (see below) and discuss thoroughly the limitations of their study.

1- Figure 1. The experiment compares gene expression between the control and the *ycg1*-degron 1 strains. The comparison before auxin treatment (at 1.5hrs) is missing. Hence, we don't know whether gene expression changes can be attributed to condensin depletion, an initial gene expression difference between the two strains, or both. The sentence (page 7, lane 9) "mRNA sequencing of the G1 sample revealed elevated expression of approximately 300 genes in the absence of condensin" is therefore incorrect because it suggests that changes are solely due to condensin depletion.

2- Fig. 5 to Fig.7. The experimental scheme (Fig. 5A) lacks controls showing that the untreated *ycg1*-degron strain is able to respond to heat-shock as the control. For instance In Fig.7B, it is clear that *YDJ1* expression is down-regulated in the control but not in the *ycg1* degron strain. However we don't not know if the effect was dependent on the Met+auxin treatment because the untreated control + HS is missing. It is difficult to rule out the possibility that the untreated degron strain is simply unable to respond properly to HS.

Minor points

Fig. S1 B. The aim was to show that both strains used in the Fig.1 experiment progressed similarly though the cell cycle. However, the control strain in Fig. S1 is different from the one used in Fig.1.

Fig. S1. E. The numbers in the Venn diagrams do not fit those in Fig.1B.

Fig.2A-C and Fig.3BC. mRNA levels were assessed by qPCR relative to *UBC6*. It is assumed that *UBC6* expression remains constant throughout the experiment but the authors provide no evidence for it.

Fig. S2A. Same comment. In addition, the control gene is now changed to *ACT1*. Why? The legend indicates *FIG1* mRNA levels whereas it's *AGA2*. Error bars are missing.

Fig.4BC. Error bars are not defined. Data supporting the choice for *ACT1* as a reference should be provided.

We would like to thank the three reviewers for their interest in our study and for their thorough and critical appraisal. We appreciate the insightful feedback and the opportunity to revise and improve our study. Please find below a point-by-point response how we have developed the manuscript in response to each of the reviewers' concerns.

Reviewer #1

The reviewer finds our manuscript to be "*the most carefully designed study to investigate condensin's role in transcription regulation so far*" and "*that the main points of this manuscript are supported by the data sufficiently*", however the reviewer also raises "*a major concern*" as well as several minor points.

Major concern

The reviewer raises an important point, that heat shock gene expression analyses typically report striking upregulation of heat shock response genes. When internally normalised, our nascent RNA-seq analysis recapitulates these **relative** gene expression changes very well. This is documented in Figure S5C, comparing our results to the reference dataset from Pat Brown's lab (Gasch et al. 2000 *Mol. Biol. Cell* 11, p4241). As an example, this includes relative upregulation of the two stress response genes shown in Figure 7A, whose absolute expression level decreases following heat shock.

In contrast, there are only a few published analyses that look at **absolute** gene expression changes following cellular stress. These include Mahat et al. 2016 *Mol. Cell* 62, 63–78, who document global transcriptional repression following heat shock, as well as Rosa-Mercado et al. 2021 *Mol. Cell* 81, p502, who report transcriptional shutdown following osmotic stress.

We do not yet know the mechanism for transcriptional repression following heat shock in budding yeast, though increased promoter-proximal RNA polymerase II pausing is thought to be responsible in human cells. We therefore completely agree with the reviewer that studying pol II occupancy following heat shock will be an important experiment in the future. For the purpose of the present study, we have performed a new experiment to alleviate the reviewer's concern that 4-TU uptake might be affected following heat shock. In our new supplementary Figure S6A, we show that 4-TU incorporation into nascent RNA is downregulated not only in response to heat shock but also in response to a wide range of other environmental changes, some as mild as the shift to a different growth medium. This confirms that global transcriptional downregulation forms part of the cellular stress response.

Minor points

- Page 7 line 2 and Fig 1C: The condensin binding sites are based on the ChIP-chip data contained in D'Ambrosio et al. 2008 *Genes Dev.* 22, p2215, which is now cited. For the distance analysis, all genes were divided into those that significantly change, or not. Distances are then recorded for all genes to their closest feature, e.g. condensin binding site. As there are many more genes than features, these distances show a broad distribution and do not simply report on the closest gene to each feature. E.g. the median distance of significantly changed genes in the G1 total RNA sequencing experiment from their nearest condensin binding site was 3,916 bp, while the median distance of unchanged genes was 8,382 bp. These distances do not by themselves carry biological meaning, but dividing one

by the other leads to the reported proximity association value of 0.47. This is now better explained, and the example is included, in the Material and Methods.

- Page 12 line 5 and Fig 5B: A list of the genes whose expression significantly changes both in the comparison of the *ycg1*^{degron2} strain before and after auxin addition, as well as between a control and *ycg1*^{degron2} strain following treatment, is now provided in a new Table S1.
- Read mapping: A new Table S3 details the mapping statistics.
- Principal component analysis: To perform the principal component analysis, the normalised counts were variance-stabilised using the *rlog* function within DESeq2. The samples were then projected onto the principal components using the standard R function *princomp* on all genes without further scaling. This is now explained in the Materials and Methods section on page 24.
- t-test: on further reflection we realise that an unpaired t-test is more appropriate to evaluate the comparison in Figure 2A. This results, as the reviewer expected, in a larger *p*-value of 0.05 that we now report.
- Page 17 line 24: The publication by Sutani et al. 2015 is indeed relevant here and has now been included. This publication has now also been cited in the introduction on human condensin binding to transcription start sites of active genes.

Reviewer #2

The reviewer finds that our “results are potentially important given the current controversy regarding the role played or not by condensin in the control of gene expression”. However, the reviewer also saw numerous weaknesses. We would like to thank reviewer 2 for the many excellent suggestions on how to further develop our study. We tried our best to capture all these suggestions and respond to each of them, below:

Major points

(1) The functionality of the degron alleles prior to auxin-mediated depletion:

The reviewer is right that the transcriptome analysis of the *ycg1*^{Degron1} strain is confounded by the fact that we compare control cells to *ycg1*^{Degron1} cells, both following medium shift and auxin treatment, without knowing the effect of the *ycg1*^{Degron1} allele before depletion. During the revisions, we have analysed the accuracy of rDNA segregation, a sensitive marker of condensin function, in control and *ycg1*^{Degron1} cells before and after depletion. This analysis is shown in the updated Fig S1C and reveals that rDNA segregation is unaffected in the *ycg1*^{Degron1} strain before depletion. This means that we can at least exclude aneuploidy as a major cause of transcriptional changes before condensin depletion. Of course, we cannot exclude more subtle effects in the *ycg1*^{Degron1} strain. Throughout the revised manuscript, we make it clear that our analysis compares two strains, not simply the consequence of condensin depletion.

The reviewer suggests that we perform gene expression analyses in *ycg1*^{Degron1} cells prior to condensin depletion. However, given that the *ycg1*^{Degron1} experiment is subject to

additional limitations, e.g. reduced Ycg1 protein stability and the medium change, we instead designed our second RNA sequencing experiment in which we analyse gene expression before and after depletion using the improved *ycg1*^{Degron2} allele and avoiding the rich medium change. We have conducted further analyses of this second experiment during the revisions, as discussed below.

(2) Spatial correlation between gene deregulation and condensin localisation

We thank the reviewer for suggesting a possible reason why the positive spatial correlation between condensin-regulated genes and condensin binding sites is stronger in case of the total RNA sequencing data, where ribosomal protein genes indeed feature more prominently amongst a smaller number of significantly changed genes. This possible explanation is included in the revised manuscript on page 8.

We have performed a similar spatial correlation analysis using the genes found to be differentially expressed following condensin depletion in the nascent RNA sequencing experiment. This has confirmed closeness of differentially expressed genes to condensin (as well as cohesin) binding sites (see the new Figure S7B).

A new Figure S7A, in turn, depicts the direction and amplitude of expression change of all ribosomal protein genes. Ribosomal gene expression is somewhat greater in the *ycg1*^{Degron2} strain already before depletion, a difference that increases by around 1.4 – 1.5-fold following depletion.

As suggested, we have analysed the statistical significance of the overlap between differentially expressed genes identified by mRNA sequencing with those detected by total RNA sequencing. A hypergeometric test showed a very strong significance of the overlap ($p = 9.1 \times 10^{-114}$ and $p = 1.0 \times 10^{-134}$ for the samples analysed in G1 and M, respectively). These p values are included in the figure legend of Figure S1.

The reviewer points to another striking correlation, that of condensin-regulated genes and cohesin binding sites. A few condensin binding sites overlap with cohesin, but the two complexes show a mostly alternating association pattern along chromosomes (D'Ambrosio et al. 2008 *Genes Dev.* 22, p2215). We take up the reviewer's observation in our revised manuscript but admit that we do not yet know the reason for the relationship of differentially expressed genes and cohesin binding sites.

(3) rDNA transcription and RPG expression

The decreased rDNA copy number indeed means that the same (or greater following condensin depletion) amount of rRNA is transcribed from a smaller number of repeats. Over the range of rDNA repeat reduction encountered in our strains (i.e. well above critical repeat tract shortening) rRNA homeostasis is thought to arise from the activation of a larger proportion of rDNA repeats, as well as from increased transcription from given repeats (French et al. 2003 *Mol. Cell. Biol.* 23, p1558). While it will be interesting to ascertain the number of active rDNA repeats in our experiments in the future, our current analysis remains limited to considering overall rRNA expression.

The *UBC6* transcript abundance, used to normalise *ITS1* expression, did not noticeably change following condensin depletion. This is shown in a new Fig S2B.

We confirmed upregulation of rRNA expression following condensin depletion in our nascent RNA sequencing experiment, which is shown in a new Fig. S7C. We thank the reviewer for pointing us to the study by Albert et al. 2016 *Mol. Cell* 64, p720, who have provided a molecular link between rRNA and ribosomal protein gene expression. Consistent

with the possibility that the Utp22-Fhl1 axis contributes to ribosomal protein gene upregulation, we find many of its components amongst the genes that are quantitatively upregulated following condensin depletion. This is shown in a new Fig. S7D and the possible implications for ribosomal protein gene regulation are discussed in the text.

(4) Transposon activation

The reviewer is right that transposon gene expression in wild type cells is increased only by a small amount (an average 1.2-fold) in rich medium. Despite being highly reproducible, this change remains below statistical significance. We have therefore adjusted the text that describes these results accordingly.

The steady state levels of *ACT1* used to normalise transposon gene expression did not noticeably change and is now provided in Figs S4A.

(5) Nascent RNA analyses using YCG1-degron2 strain.

(i) The median change amplitude of the significantly changed group of genes was around 1.4-fold. Exact numbers for both up and down-regulated genes in both the *ycg1*^{Degron2} strain before and after auxin addition, as well as between control and *ycg1*^{Degron2} strain after auxin addition, are now found in the Figure 5 legend.

(ii) We have also performed a gene ontology analysis of the genes in common between both above comparisons, which revealed a large number of genes involved in the ribosome biogenesis pathway, consistent with the idea that ribosome biogenesis is affected in coordination with increased rRNA expression. This analysis is contained in a new Figure 5C.

(iii) Spatial distance analysis of the genes in common between both comparisons again showed a strong correlation with condensin (as well as cohesin) binding sites, shown in a new Figure S7B. This correlation is lost following heat shock, when of course the vast majority of genes show differential behaviour between the control and *ycg1*^{Degron2} strain.

We provide a table of both the genes in common between Figure 1 and Figure 5, as well as between the two comparisons in Figure 5, in a new Tables S1 for the reader's perusal.

The effect of condensin depletion following auxin treatment and following heat shock

The reviewer asks for clarification of the effect of condensin depletion, compared to the additional consequences of heat shock. The reviewer is right that methionine and auxin addition caused a mild downregulation of gene expression in the control, but not in the condensin depletion strain, which is made clear in the revised manuscript. Following heat shock, gene expression is downregulated in both control and condensin depletion strains. The fold change of the downregulation is substantially greater in the control strain (2.5-fold), which is reduced following condensin depletion to 1.6-fold. This part of the manuscript has been rewritten to provide a clearer description.

Genes upregulated in Figure 6A and 6C

Figure 6C counts only genes whose expression differences are statistically significant and are greater than 2-fold. Figure 6A visualises all changes, including those of genes that change to a lesser degree.

(6) The nascent transcription of highly expressed genes.

We indeed observe elevated levels of ribosomal protein genes in the *ycg1*^{Degon2} strain before methionine and auxin addition. However, following treatment, ribosomal protein gene expression is further elevated. This observation is now better explained and the gene expression changes of all ribosomal protein genes, both before and after methionine and auxin addition, can be seen in a new Fig S7A. This representation confirms increased ribosomal protein gene expression across all gene family members.

rRNA expression

Based on the reviewer's excellent suggestion, we have looked at rDNA expression in our nascent RNA sequencing experiment. Consistent with the observations in the total RNA sequencing experiment (Fig 2C), *RDN37* expression in the nascent RNA sequencing experiment increased following condensin depletion. This was the case both when comparing the *ycg1*^{Degon2} strain before and after depletion or when comparing the control and *ycg1*^{Degon2} strain after methionine and auxin addition. These results are included in a new Fig S7C.

A role of the Utp22/Ifh1 axis in ribosomal protein gene expression changes

Following the reviewer's very pertinent suggestion, and having confirmed increased rRNA transcription in the absence of condensin, we considered the possible role of the Utp22-Ifh1 axis. Indeed *RRP7*, *IFH1*, *RAP1*, *UTP22* and *FHL1* were all amongst the genes whose expression was quantitatively upregulated following condensin depletion. Details of the changes can be found in the new Fig S7D.

(7) tRNA gene expression

In response to the reviewer's interesting suggestion, we explored tRNA expression and its response to condensin depletion. We indeed see greater read numbers at tRNA genes following condensin depletion. However, overall read numbers were small. The observed increases therefore reached statistical significance between the three experimental repeats only at two tRNA loci. A likely reason is that our RNA extraction and sequencing protocols were not designed to capture these short transcripts with strong secondary structures. We have to defer a more thorough analysis of tRNA expression and its possible regulation by condensin to future investigations.

Minor points

- *Introduction, page 3.*

D'Ambrosio et al. 2006 *Genes Dev.* 22, p2215 saw a strong correlation between condensin binding sites and sites occupied by the Scc2-Scc4 cohesin loader. At the time, the identity of binding sites other than tRNA and RP genes was unknown. Lopez-Serra et al. 2014 *Nat. Genet.* 46, p1147 then found that the remaining Scc2-Scc4 binding sites, and by inference condensin binding sites, correspond to promoters of other highly expressed genes. This is clarified. Similarly, *S. pombe* and human condensin (Sutani et al. 2015 *Nature Comm.* 6, 7815 and *C. elegans* condensin (Kranz et al. 2013 *Genome Biol.* 14, pR112) are found at transcription start sites of highly expressed genes.

- *Fig. 1B vs Fig. S1.*

Fig 1B shows genes differentially expressed by 1.5-fold or greater, while Fig S1E is made up of a greater number of genes that show significantly different expression levels between the

two experimental strains without requirement for a 1.5-fold threshold. We have updated the Fig S1E legend to make this clear.

- *Figure S2 Legend.*

Apologies for the mix-up, *FIG1* mRNA levels are actually shown in Figure 2A. Figure S2 has now been repurposed to show expression changes of all ribosomal protein genes as well as the absolute expression levels of *UBC6* and *ITS1*.

- *Fig. S5A.*

We thank the reviewer for picking up this labelling mistake, which has been corrected.

- *Statistical significance in Fig. 5B, Fig. S5B-C.*

p values for the significance of overlaps based on hypergeometric tests have been included in the figure legends of all Venn diagrams.

- *could the authors discuss why a difference in gene expression upon heat shock has not been detected by Hocquet et al. 2018 or Nakazawa et al. 2015*

Hocquet et al. 2018 used condensin inactivation for 2.5 hours at 36 degrees, after which the acute effect of heat shock might have relinquished. Nakazawa et al. 2015 in turn show an expression timecourse of heat shock-induced genes, internally normalised to Actin. Looking at the published primary data, heat shock gene expression appears somewhat augmented following condensin inactivation. This could agree with our observation that absolute downregulation of these genes following heat shock is less efficient in the absence of condensin.

Reviewer #3

The reviewer finds that “*A role for condensin in facilitating transcriptional adaptation [...] is appealing and would be of great interest to a large audience*”, but also raises “*concerns about the experimental system that cast doubts about the main conclusions*”

Main points

1- Figure 1. “mRNA sequencing of the G1 sample revealed elevated expression of approximately 300 genes in the absence of condensin”

We thank the reviewer for pointing out that this sentence is misleading. We have reworded the sentence to read “mRNA sequencing of the G1 sample revealed higher expression of approximately 300 genes in the strain depleted for condensin, when compared to the control strain...”. Throughout the text, we now refer to gene expression differences between strains rather than to consequences of condensin depletion. We also added a paragraph to discuss the limitation of our approach at the end of the first results section on page 8.

We should mention that we have designed the second, nascent RNA sequencing experiment specifically to address the reviewer’s concern. In this second experiment, we compare the control and condensin depletion strains both before and after methionine and auxin addition.

2- Fig. 5 to Fig.7. The experimental scheme lacks controls showing that the untreated *ycg1*-degron strain is able to respond to heat-shock as the control.

The untreated degron strain differs from the otherwise isogenic control strain by two modifications to the *YCG1* locus, a repressible promoter and the degron tag. During our revisions, we have analysed the fidelity of rDNA segregation during cell division in the *ycg1*^{Degon2} strain, a sensitive assay to evaluate condensin function. This analysis is included in the revised Fig S1C and shows that rDNA segregation occurs with equal fidelity to control cells before methionine and auxin addition. This suggests that *ycg1*^{Degon2} cells contain functional condensin before depletion. Of course, we cannot exclude more subtle defects that are not picked up by this assay. This is now made clear in the revised manuscript on page 11.

Minor points

Fig. S1B

The reviewer is right that the control strain in Fig S1B was different from the one used in Fig 1. During the revisions, we have repeated the experiment shown in Fig S1B with the exact strains that were used in Figure 1. The result from this new experiment is now shown in Fig S1B.

Fig. S1E

We thank the reviewer for pointing out the discrepancy in the numbers between the two figure panels. Fig 1B shows all genes significantly differentially expressed 1.5-fold or greater, while Fig S1E shows all significantly changed genes between the control and condensin depletion strains without a required 1.5-fold threshold. This results in their greater number. We have revised the Fig S1E legend to clarify this point.

Fig.2A-C and Fig3BC

Levels of *UBC6* and *ITS1* transcripts detected by qPCR, relative to a standard curve of commercial *S. cerevisiae* DNA, as well as of *UBC6* and the various histone genes, are now shown prior to normalisation in Fig S2B and Figs S3B and D, respectively. This confirms that *UBC6* levels remained constant throughout the experiment.

Fig. S2A

This Figure has now been repurposed to show *UBC6* and *ITS1* transcript levels before normalisation. The *AGA2* analysis, that was peripheral to our study and additional to the *FIG1* analysis in Fig 2A, was removed.

Fig. 4BC

We thank the reviewer for pointing out this omission. The error bars represent standard deviation and this has now been indicated in the figure legend. *ACT1* is a historically frequently used control in gene expression studies, though more recently *UBC6* has been advocated to serve as a more robust control. Where we use *ACT1* as the reference for transposon gene expression in Fig 4, we have now also added the measurements of transcript levels before normalisation in a new Fig S4C.

May 19, 2021

RE: Life Science Alliance Manuscript #LSA-2020-00961R

Dr. Frank Uhlmann
The Francis Crick Institute
Chromosome Segregation Laboratory
1 Midland Road
London NW1 1AT
United Kingdom

Dear Dr. Uhlmann,

Thank you for submitting your revised manuscript entitled "A role for condensin in mediating transcriptional adaptation to environmental stimuli". We believe the paper is developed sufficiently to be published in Life Science Alliance pending minor concerns raised by Reviewer 1 and 2, one experimental request raised by Reviewer 3 and final revisions necessary to meet our formatting guidelines.

We apologize for this delay in getting back to you. We are happy to see that revisions have improved the manuscript, a point that is also clear in the reviewers' comments (see below). There is just one exceptional issue remaining that has been outlined by Reviewer 3 in their comments. We encourage you to check a few genes in response to heat shock in the absence of auxin or alternatively, tell us why they wish not to do this in a point-by-point response.

In the interest of shortening the time to publication, we also encourage you to address the following edits in the revised manuscript to meet our formatting guidelines,

- please upload your main and supplementary figures as single files
- please add your main, supplementary, and table legends to the main manuscript text after the reference section
- please add callouts for Figures 5C, S7C, and D to your main manuscript text
- please add scale bar for Figure S1C
- please provide a higher quality image for Figure 5A
- please provide blots from duplicate experiments shown in Figure S2C

A. FINAL FILES:

B. MANUSCRIPT ORGANIZATION AND FORMATTING:

Sincerely,

Shachi Bhatt, Ph.D.
Executive Editor
Life Science Alliance
<http://www.lsjournal.org>
Tweet @SciBhatt @LSAJournal

Reviewer #1 (Comments to the Authors (Required)):

In the revised manuscript, the authors have fully resolved the concerns I had in the first manuscript. The text has been carefully rewritten so that it conveys the experimental results more clearly and accurately. Their relevance to the literature is adequately discussed. The methods section and legends provide sufficient information to repeat the experiments and analyses. I think that the current manuscript is appropriate for publication in Life Science Alliance.

One minor comment to the authors: please double check if the "mean"s in the first paragraph of page 14 is actually mean and not median. It seems inconsistent with Fig. 6B, where the median is depicted.

Reviewer #2 (Comments to the Authors (Required)):

Lucy Lancaster and colleagues have addressed most of my comments, and I consider that their manuscript is now suitable for publication in LSA. I have one remaining request that I consider important for the clarity of the message delivered to the reader and noticed few typos.

Request:

Page 8, related to the sentence: "While the fidelity of rDNA segregation, a sensitive readout for condensin function, was unaffected before depletion (Fig S1C), we cannot exclude that gene expression differences existed between the control and ycg1Degron1 strain already before condensin depletion."

I acknowledge the verification of rDNA segregation with respect to the question of aneuploidy as a possible driving force in transcriptome changes, and the formal word of caution. However, I think that the authors could go a bit further in that matter. The authors clearly show a marked reduction in the number of rDNA repeats in the ycg1Degron1 strain before depletion. This phenotype, though not coupled with mitotic missegregation, argues that condensin is impaired in the ycg1Degron1 strain already before depletion. Thus, I suggest for the sake of the readers that the authors mention this important conclusion.

Typos

Page 8: We confirmed that gene expression differences seen in our mRNA sequencing data could be independently reproduced by quantitative real-time PCR (qPCR) analysis. Fig 2A shows an example of the pheromone response gene FIG1, that showed higher expression in condensin-depleted G1 cells.

Can the authors mention that UBC6 mRNA steady state levels are shown in Fig S2B or S3B.

Page 10: This analysis confirmed that indeed Ty2, but not Ty1, expression was almost two-fold lower in condensin-depleted G1 cells when compared to control cells (Figs 4B and S4B).

Typo? Fig. S4A instead of S4B?

Page 15: Consistent with our results obtained by total RNA sequencing, we found that synthesis rates of the large 35S rRNA transcript, RDN37 produced by RNA polymerase I, increased around 1.5-fold following condensin depletion. Additionally, several genes of the Utp22-lfh1 axis were amongst those that were significantly more strongly expressed in the ycg1Degron2 strain, compared to the control, following condensin depletion. References to Fig S7C, S7D are missing.

Reviewer #3 (Comments to the Authors (Required)):

In their revised manuscript the authors partially addressed the main points I raised. However a specific concern remains about point 2 that I recall here:
"Fig. 5 to Fig.7. The experimental scheme (Fig. 5A) lacks controls showing that the untreated ycg1-degdon strain is able to respond to heat-shock as the control. For instance In Fig.7B, it is clear that YDJ1 expression is down-regulated in the control but not in the ycg1 degdon strain. However we don't not know if the effect was dependent on the Met+auxin treatment because the untreated control + HS is missing. It is difficult to rule out the possibility that the untreated degdon strain is simply unable to respond properly to HS."

The authors did not address directly the question. Instead, they analyzed the fidelity of rDNA segregation in the ycg1Degon2 strain (revised Fig S1C) and found that rDNA segregation occurs with equal fidelity to control cells before methionine and auxin addition. This suggests that "ycg1Degon2 cells contain functional condensin before depletion". To me, this solely indicates that the untreated strain retains sufficient condensin function to properly segregate its DNA. Still, the question raised in point 2 has not been addressed: is the untreated degdon strain able to respond properly to heat shock? If not, this would strongly suggest that the effect they describe may be an indirect consequence of a constitutively altered condensin function. I would have been convinced if the authors had provided data showing that the untreated degdon strain did show a global transcriptional shut-down in response to HS. At minima, they could have monitored the expression of several genes, as in Fig. 7B.

This is an important issue because that condensin mediates transcriptional adaptation to environmental stimuli is the main claim of the paper (i.e., the title).

We would like to thank the three reviewers for their valuable additional feedback on our revised manuscript. Please find below a point-by-point response how we have made additional changes to the manuscript in response to the reviewers' suggestions.

Reviewer #1

We thank the reviewer for pointing out an inconsistency between what is described in the text and what is highlighted in Fig 6B. The text reports on the mean read count ratios, while medians are highlighted in the figure. Given the close range of the datapoints, there is little difference between the means and the medians. Nevertheless, for consistency between the text and the figure, we now report on the median read count ratios in the text. This introduces marginal changes to the reported values. It does not alter any conclusion.

Reviewer #2

Request:

The reviewer mentions the possibility that altered rDNA repeat numbers in the *ycg1*^{Degron1} strain might be behind some of the observed gene expression differences. This is a fair point that we have taken up. At the end of the paragraph that describes rDNA repeat number changes on page 9, we add: "We cannot exclude that the rDNA copy number change has indirectly contributed to gene expression differences observed between the control and *ycg1*^{Degron1} strains."

Typos:

Page 8: A reference to Fig S2B, showing that *UBC6* mRNA steady state levels are constant, has been added to the Figure 2A legend.

Page 10: Quantitative transposon transcription confirmation is indeed reported in Fig S4A, not Fig S4B. We thank the reviewer for spotting this mistake, which has been corrected.

Page 15: References to Figs S7C and S7D have now been added on pages 15/16.

Reviewer #3

The reviewer requests that we analyse gene expression changes following heat shock in the *ycg1*^{Degron2} strain, before and after condensin depletion.

As always with a complex experimental system, we have to make a choice as to which comparisons we make and how we control for the effects that we see. In this case, careful deliberations led us to the design of our heat shock experiment. We use two yeast strains that are isogenic apart from the respective *YCG1* or *ycg1*^{Degron2} alleles. This pair of strains is then experimentally identically treated, allowing us to conclude on the impact of the *ycg1*^{Degron2} allele on the heat shock response. A limitation of this comparison is that, despite our controls showing functionality of the *ycg1*^{Degron2} allele before depletion, we cannot be completely sure whether the *ycg1*^{Degron2} allele might have affected the heat shock response even before inactivation.

To circumvent this, the reviewer suggests an alternative experimental design. We can use the *ycg1*^{Degron2} strain and compare its heat shock response before and after the

condensin depletion treatment. This design will reveal whether the *ycg1*^{Degron2} allele affects the heat shock response already before depletion. The drawback of this design is that we then do not know the origin of any changes that result from treatment. These could be indeed a consequence of condensin depletion, or they could be an indirect consequence of the treatment.

Both experimental designs therefore have their advantages and disadvantages. In our view, if there was an effect of the *ycg1*^{Degron2} allele already before depletion, it would likely stem from a partial loss of function. The experiment suggested by the reviewer would therefore capture a weakened version of the response that follows depletion. For this reason, we chose an experimental design that allows us to compare the full wild type function of *YCG1* to its full depletion phenotype.

Given the time requirements that a new nascent RNA sequencing experiment and its downstream data analysis would require, it is unfortunately beyond the scope of this present study to repeat the heat shock analysis using an alternative design.

May 21, 2021

RE: Life Science Alliance Manuscript #LSA-2020-00961RR

Dr. Frank Uhlmann
The Francis Crick Institute
Chromosome Segregation Laboratory
1 Midland Road
London NW1 1AT
United Kingdom

Dear Dr. Uhlmann,

Thank you for submitting your Research Article entitled "A role for condensin in mediating transcriptional adaptation to environmental stimuli". It is a pleasure to let you know that your manuscript is now accepted for publication in Life Science Alliance. Congratulations on this interesting work.

DISTRIBUTION OF MATERIALS:

Again, congratulations on a very nice paper. I hope you found the review process to be constructive and are pleased with how the manuscript was handled editorially. We look forward to future exciting submissions from your lab.

Sincerely,

Shachi Bhatt, Ph.D.

Executive Editor

Life Science Alliance

<http://www.lsajournal.org>
